# Repurposing cancer drugs identifies kenpaullone which ameliorates pathologic pain in preclinical models via normalization of inhibitory neurotransmission

Michele Yeo [1,7 ✉], Yong Chen [1,7 ✉], Changyu Jiang[2], Gang Chen[2], Kaiyuan Wang[2], Sharat Chandra[2], Andrey Bortsov[2], Maria Lioudyno[3], Qian Zeng[1], Peng Wang[1], Zilong Wang [1,2], Jorge Busciglio[3], Ru-Rong Ji [2,4 ✉] & Wolfgang Liedtke [1,2,4,5,6 ✉]

Inhibitory GABA-ergic neurotransmission is fundamental for the adult vertebrate central nervous system and requires low chloride concentration in neurons, maintained by KCC2, a neuroprotective ion transporter that extrudes intracellular neuronal chloride. To identify *Kcc2* gene expression-enhancing compounds, we screened 1057 cell growth-regulating compounds in cultured primary cortical neurons. We identified kenpaullone (KP), which enhanced *Kcc2/KCC2* expression and function in cultured rodent and human neurons by inhibiting GSK3ß. KP effectively reduced pathologic pain-like behavior in mouse models of nerve injury and bone cancer. In a nerve-injury pain model, KP restored *Kcc2* expression and GABA-evoked chloride reversal potential in the spinal cord dorsal horn. Delta-catenin, a phosphorylation-target of GSK3ß in neurons, activated the *Kcc2* promoter via KAISO transcription factor. Transient spinal over-expression of delta-catenin mimicked KP analgesia. Our findings of a newly repurposed compound and a novel, genetically-encoded mechanism that each enhance *Kcc2* gene expression enable us to re-normalize disrupted inhibitory neurotransmission through genetic re-programming.

---

[1] Department of Neurology, Duke University Medical Center, Durham, NC, USA. [2] Department of Anesthesiology (Center for Translational Pain Medicine), Duke University Medical Center, Durham, NC, USA. [3] Department of Neurobiology & Behavior, Institute for Memory Impairments and Neurological Disorders (iMIND), Center for the Neurobiology of Learning and Memory, University of California at Irvine, Irvine, CA, USA. [4] Department of Neurobiology, Duke University Medical Center, Durham, NC, USA. [5] Duke Neurology Clinics for Headache, Head-Pain and Trigeminal Sensory Disorders, Duke University Medical Center, Durham, NC, USA. [6] Duke Anesthesiology Clinics for Innovative Pain Therapy, Duke University Medical Center, Durham, NC, USA. [7] These authors contributed equally: Michele Yeo, Yong Chen. ✉email: myeo@duke.edu; yong.chen@duke.edu; ru-rong.ji@duke.edu; wolfgang.liedtke@regeneron.com

In the mature vertebrate central nervous system (CNS), γ-aminobutyric acid (GABA) acts primarily as an inhibitory neurotransmitter and is critical for normal CNS functioning[1,2]. In chronic pain, GABA-ergic transmission is compromised, causing circuit malfunction and disrupting inhibitory neural networks[3–12]. Therapeutic approaches for restoring physiologic GABA-ergic transmission would enable us to address the unmet medical need of chronic pain with safer and more effective alternatives to opioids.

In the adult vertebrate CNS, the $K^+/Cl^-$ cotransporter KCC2 is expressed exclusively in neurons. KCC2 continuously extrudes chloride ions, thus ensuring that intracellular levels of chloride remain low, as required for inhibitory GABA-ergic neurotransmission[13–19]. In chronic pathologic pain, KCC2 expression is attenuated in the primary sensory gate in spinal cord dorsal horn (SCDH) neurons. This key pathophysiological mechanism contributes to an imbalance of excitation/inhibition because it corrupts inhibitory neurotransmission, leading to inhibitory circuit malfunction[5,7,11,20–24]. Notably, there is no 'back-up' protein that can rescue the KCC2 expression deficit. Thus, we reasoned that if we could boost *Kcc2/KCC2* gene expression (*Kcc2*—rodent gene; *KCC2*—human gene; KCC2—protein), we could re-normalize inhibitory transmission for relief of chronic pain.

We, therefore, conducted an unbiased screen of cell growth-regulating compounds. We searched among these compounds because we assumed that a sizable number of them function by interfering with epigenetic and transcriptional machinery to inhibit cell division. Since mature neurons do not divide, these compounds are attractive candidates to upregulate gene expression of *Kcc2/KCC2* via epigenetic mechanisms, thus lowering intraneuronal chloride levels and re-establishing normal GABA-ergic inhibitory functioning.

Candidate compounds were identified by rigorous iterations of primary and secondary screening leveraging metrics of *Kcc2* gene expression, seeking enhancing compounds, and we selected one for in-depth exploration: kenpaullone (KP), a glycogen synthase kinase-3 (GSK3)/cyclin-dependent kinase (CDK) inhibitor[25,26]. We subsequently found that KP functioned as an analgesic in preclinical mouse models of pathologic pain. The results of in vitro and in vivo studies suggest that the cellular mechanism of action of KP in neurons is based on its GSK3ß-inhibitory function which enhanced *Kcc2/KCC2* gene expression. *Kcc2* up-regulation, in turn, relied on the nuclear transfer of the neuronal catenin, δ-catenin (δ-cat)[27,28] which enhanced *Kcc2* gene expression via KAISO transcription factors[29]. As expected, increased *Kcc2* gene expression led to increased chloride extrusion by KCC2 transporter in neurons. Importantly, after KP treatment, patch-clamp recordings revealed more negative, thus electrically more stable GABA-evoked chloride reversal potentials in SCDH pain relay neurons of mice subjected to nerve constriction injury. In vivo, these mice showed robust analgesia in response to KP, and defective expression of KCC2 in the SCDH was repaired by KP.

## Results

**A screen of 1057 compounds in primary cortical neurons for *Kcc2* gene expression-enhancers identifies KP.** To identify *Kcc2* gene expression-enhancing compounds, we cultured primary cortical neurons from *Kcc2*-luciferase (LUC)-knockin (*Kcc2*-LUCki) mice[30,31] and used LUC as a readout for the activity of the proximal *Kcc2* promoter (2.5 kB[18]), which drives LUC in this transgenic mouse line. In our screening assay, we recorded coefficients of variation of 6% for vehicle control, and 5.3% for positive control, with further details supporting the robustness of

the assay shown in Supplementary Fig. 1a, b. We remain aware that this strategy will not select long-range enhancers of *Kcc2* gene expression that act outside the 2.5 kB core *Kcc2* promoter. We screened 1057 compounds, contained in two NCI libraries (Fig. 1, Supplementary Fig. 1b, Supplementary Data File S1), related to inhibition of growth of malignantly transformed cells. The rationale was to have a starting pool of compounds that can upregulate *Kcc2* gene expression in non-dividing neurons via epigenetic mechanisms. Iterative screening followed by measurements of *Kcc2* mRNA (RT-qPCR[18]) and intracellular chloride ([Cl⁻]i) (clomeleon chloride indicator protein[18,32]), led to decreasing numbers of re-confirmed hits, and we ended up with 4 "winner" compounds (Fig. 1). Of these, we identified KP, a GSK3/CDK kinase inhibitor, as a promising compound for further study based on its previous record of neuroprotection in translationally relevant preclinical models[33–36]. KP evoked increased activity of the *Kcc2* promoter starting with 10 nM and saturating at 1000 nM (Supplementary Fig. 1c), with an estimated $EC_{50}$ of 90 nM, which is close to recorded biochemical EC50 measurements (Supplementary Fig. 1d)[37]. As a yield of our screen, two additional "winner" compounds significantly enhanced *Kcc2* gene expression at sub-micromolar concentrations (Supplementary Fig. 1e).

We characterized KP further in primary cortical neurons. Our data establish that (i) KP enhanced *Kcc2* gene expression in rat and mouse primary cortical neurons (Fig. 2a, Supplementary Fig. 1c, d, Supplementary Data File S1), (ii) this effect of KP was dose-dependent when tested in mouse and rat neurons (10–1000 nM, Supplementary Fig. 1c, d; 100–500 nM; Fig. 2a), (iii) KP (0.5 μM) increased KCC2 protein expression with

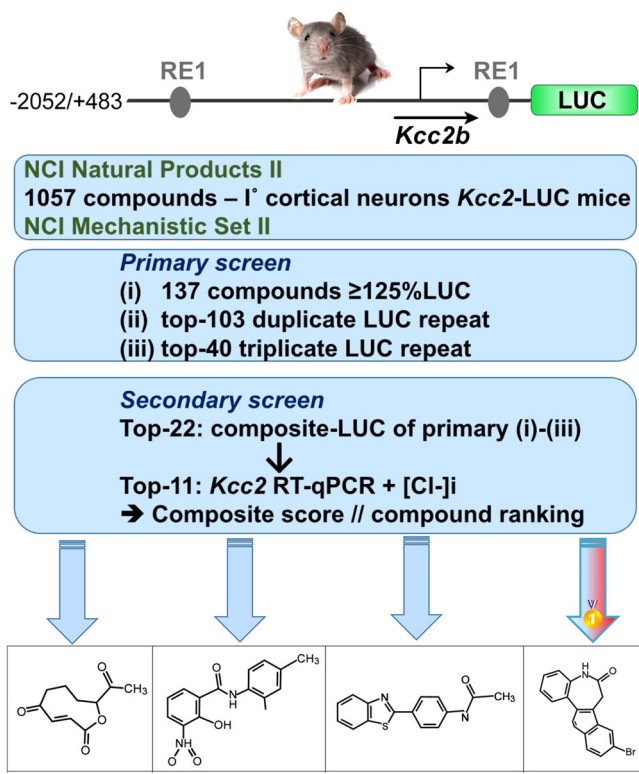

**Fig. 1 Compound screening for enhancers of *Kcc2* expression in primary cortical neurons yields Kenpaullone.** Screening paradigm using three rounds of primary screen based on luciferase (LUC) activity followed by secondary screen including *Kcc2* RT-qPCR and clomeleon chloride imaging. Bottom panel: Four compound "winners" including Kenpaullone (KP); screening conducted in primary mouse neurons.

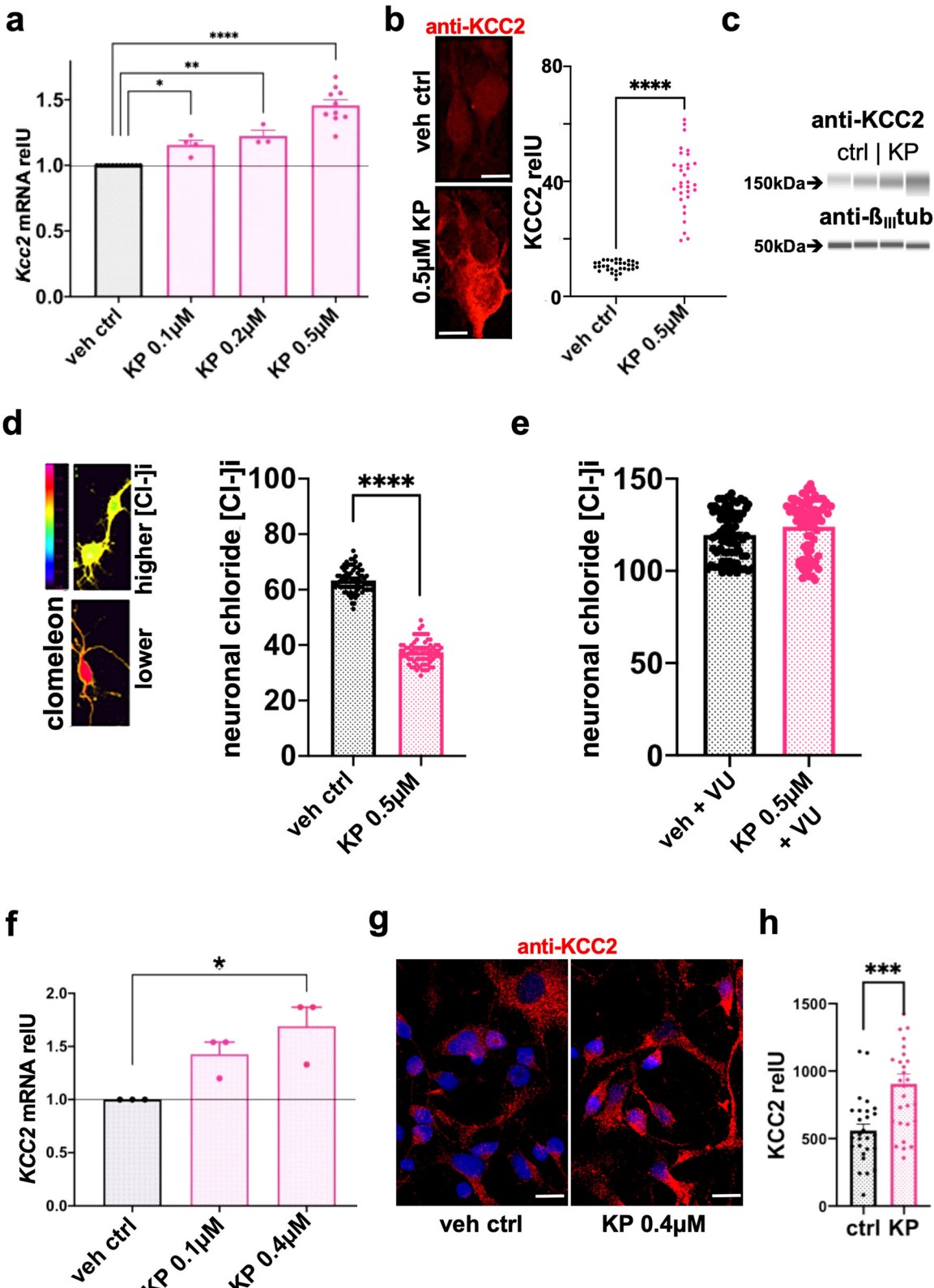

statistically significant difference vs vehicle control in rat primary cortical neurons when measured by quantitative immunocyto-chemistry (ICC) and by microcapillary size separation followed by immunodetection (Fig. 2b, c, Supplementary Fig. 2a–c), (iv) in rat primary cortical neurons, KP lowered [Cl$^-$]i with statistically significant difference vs. vehicle control (Fig. 2d), (v) this effect critically depended on chloride-extruding function of KCC2

transporter protein (Fig. 2e), (vi) KP did not function as a direct enhancer of KCC2 transporter-mediated chloride efflux (Supple-mentary Fig. 2d), (vii) KP enhanced *Kcc2* gene expression and KCC2 chloride extrusion function with rather rapid kinetics within 3 h (Supplementary Fig. 2e, f); (viii) in human primary fetal cortical neurons, KP dose-dependently enhanced *KCC2* mRNA expression (Fig. 2f). The latter finding was accompanied

**Fig. 2 Kenpaullone enhances *Kcc2/KCC2* gene expression and function in rat and human primary cortical neurons. a** Primary rat cortical neurons. Bar diagram: dose-dependent increase in *Kcc2* mRNA expression after KP treatment, which was started at DIV5, with cells harvested on DIV8. Results represent the average mRNA expression of multiple independent neuronal cultures, $n = 12$ (control), $n = 4$ (0.1 μM KP), $n = 3$ (0.2 μM KP), $n = 10$ (0.5 μM KP). Data are represented as mean values ± SEM. *$p = 0.0135$, **$p = 0.0015$, ****$p < 0.0001$, one-way ANOVA. **b** Primary rat cortical neurons. Increased protein expression as detected by KCC2 immunocytochemistry (ICC). Left-hand micrographs show representative ICC (scale bar = 10 μm), right-hand point-cloud diagram shows significantly increased densitometric measurements when treating with KP (0.5 μM; otherwise as in (**a**)); $n = 32$ neurons for ctrl, $n = 31$ neurons for KP, neurons derived from 3 independent cultures, $n = 10$–12 neurons per culture. See Supplementary Fig. 2a for validation of antibody used. ****$p < 0.0001$, two-sided *t*-test. **c** Primary rat cortical neurons. Increased protein expression as measured by KCC2 immunodetection after microcapillary electrophoretic separation of proteins from 2 independent cultures, chemiluminescent signal converted into an optical density in a virtual Western blot, as in refs. [90, 91]. Note increased expression of KCC2 in KP-treated cultures, in the 2 right-hand lanes (treatment as in (**a**), (**b**)), the lower figure shows results for detection of ß$_{III}$-tubulin, indicating similar loading with neuronal proteins. See Supplementary Fig. 2b, c, also for quantification of the KCC2 signal normalized for ß$_{III}$-tubulin. **d** Neuronal [Cl⁻]i, measured with ratiometric chloride indicator, clomeleon, is robustly and significantly reduced after KP treatment (0.5 μM, DIV5–8). Data are represented as mean values ± SEM. $n = 76$ neurons (vehicle), $n = 75$ neurons (KP)/3 independent cultures; ****$p < 0.0001$, two-sided *t*-test. **e** Note that add-on treatment with KCC2-transport blocker, VU0240551 (2.5 μM), leads to a [Cl⁻]i ≥ 120 mM, for both, vehicle-treated and KP-treated, indicating that KP's chloride lowering effect relies on KCC2 chloride extruding transport function. Data are represented as mean values ± SEM. $n = 76$ neurons (both groups)/3 independent cultures. **f** Primary human fetal cortical neurons. *KCC2* mRNA increases in a dose-dependent manner upon treatment with KP, treatment applied DIV6 to DIV8. Data are represented as mean values ± SEM of mRNA expression, three independent neuronal cultures. Data are represented as mean values ± SEM. *$p = 0.013$, one-way ANOVA. **g** Primary human fetal cortical neurons; treatment with 0.4 μM KP, as in (**f**). Representative confocal images at DIV10 immuno-labelled for KCC2, based on three independent neuronal cultures resulting in a total of 27 confocal slices for vehicle control and 28 confocal slices for KP-treated. Note the enhanced expression of KCC2 in response to KP, recapitulating findings in rat neurons (Fig. 2b). Scale bar = 10 μm. **h** Primary human fetal cortical neurons. Morphometry of KCC2 ICC shows significantly increased KCC2 expression (62%) vs vehicle after KP treatment (0.4 μM, as in (**f**)). Data are represented as mean values ± SEM. $n = 27$ (vehicle control), $n = 28$ (KP) confocal slices harboring 20–30 neurons per slice; ***$p = 0.0002$, two-sided *t*-test. Source data are provided as source data files.

by increased protein expression of KCC2 (Fig. 2g), with both markers of *KCC2* gene expression increased in a statistically significant manner vs. vehicle control. Importantly, this finding adds translational-medical relevance to the new concept of KP as a *Kcc2/KCC2* gene expression-enhancer. Moreover, we recorded significantly increased expression of the neuronal maturation marker, synaptophysin, which colocalized with KCC2 (Supplementary Fig. 2g, h).

These findings in rodent and human neurons indicate that our rationally designed screen identified a GSK3/CDK kinase inhibitor, KP, that enhanced *Kcc2/KCC2* gene expression while having no direct effect on KCC2-mediated chloride extrusion in CNS neurons. Thus, KP functioned as *Kcc2/KCC2* gene expression-enhancer in mammals including humans.

**KP functions as an analgesic in vivo**. In view of these findings, we decided to address the in vivo analgesic effects of systemically applied KP. For this, we used two types of preclinical mouse models of pathologic pain: peripheral nerve constriction-induced neuropathic pain, using the PSNL method of nerve constriction injury[38,39], and a bone cancer pain model that relies on the implantation of mouse lung carcinoma cells into the marrow of the femur[40–42]. Nerve constriction injury, a widely used neuropathic pain model, was previously used to demonstrate down-regulated KCC2 expression in the SCDH in pain[3,6]. Bone cancer pain is another clinically relevant pain model. In particular, it fits the profile of KP, as the compound inhibits GSK3 and CDKs. These actions might possibly be analgesic but also antineoplastic toward the implanted carcinoma cells.

We found that behavioral sensitization (mechanical allodynia) in both pain models, namely a decrease in withdrawal thresholds in response to mechanical cues, was significantly reduced by i.p. KP treatment, as illustrated in Fig. 3a, b. Analgesic effects of KP appeared to be dose-dependent in both pain models. In nerve constriction, KP 10 mg/kg daily intraperitoneal (i.p.) injections were effective starting d7, and significantly more effective at 30 mg/kg (Fig. 3a, Supplementary Fig. 3a). Importantly, we observed delayed onset analgesia of long duration, e.g. an almost complete elimination of pain hypersensitivity on d7–14. In bone

cancer pain, a significant analgesic effect was seen only at 30 mg/kg (Fig. 3b). The analgesic effect in this model was similarly characterized by delayed onset and prolonged duration, becoming apparent at d10 and d14. This protracted time-course of analgesic activity in both pain models suggests re-programming of gene expression that underlies pain hypersensitivity, which is characteristic of epigenetic gene regulation[43,44]. In contrast, our findings do not suggest inhibition of a nociceptive ion channel or receptor because the onset of analgesic action would be less protracted. In summary, KP functions as an analgesic in vivo when administered systemically, both in a neuropathic pain model and in a bone cancer pain model. Of note and in contrast to its effective analgesic profile in bone cancer pain, KP did not significantly inhibit osteolysis/bone damage (Supplementary Fig. 3b), indicating that KP's analgesic effect was not a result of limiting the growth of the implanted mouse lung carcinoma cells.

We then asked whether the analgesic effects of KP were mediated centrally at the spinal cord level. We injected KP intrathecally (i.t.; 30 μg, daily for 3 days) and observed reduced mechanical allodynia in mice with nerve constriction injury, using the CCI method of nerve constriction (Fig. 3c). The effect was apparent at 3 h after dosing and persisted for 24 h. This time-course is in keeping with the rapid effect on *Kcc2* gene regulation that we observed in primary cortical neurons (Supplementary Fig. 2e, f). Interestingly, we found an accumulating effect of KP after the 2nd injection, and reduction of mechanical allodynia was statistically significant at both 3 and 24 h (Fig. 3c). In order to learn whether the i.t. analgesic effect of KP relies on KCC2 transporter-function, we next co-applied KP with the KCC2 chloride transport inhibitor VU0240551 (30 μg i.t.) and observed elimination of the i.t. analgesic effects of KP for several hours (Fig. 3d). This observation suggests that the central analgesic effect of KP depended on KCC2-mediated chloride extrusion.

In view of these beneficial effects of KP on pathologic pain, and to prepare for translation toward clinical use, we next addressed whether KP had undesirable effects on the CNS in terms of sedation, impairment of motor stamina, balance, and coordination. Rotarod testing[45] of KP-treated mice (10, 30 mg/kg; i.p.) showed that KP did not induce such unwanted side effects

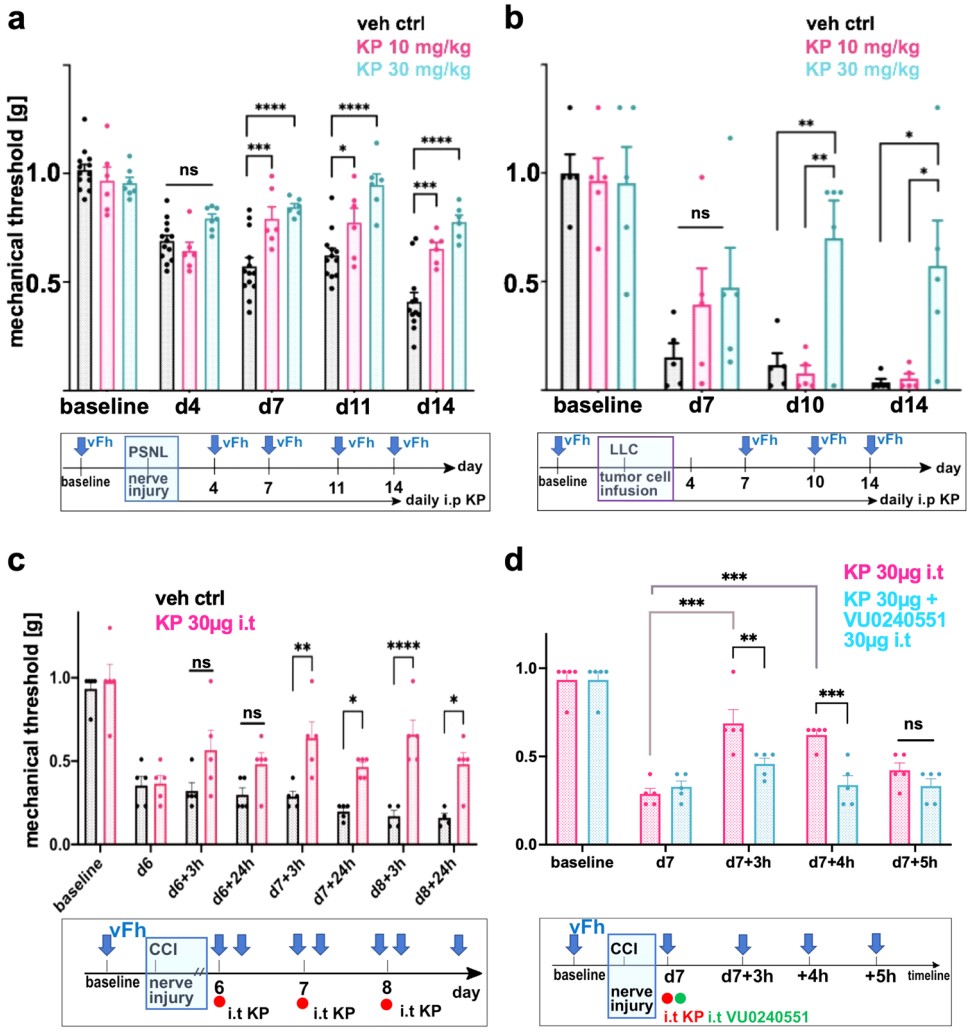

**Fig. 3 Kenpaullone is an analgesic in mouse nerve constriction injury and bone cancer pain. a** Systemic KP is an analgesic for nerve constriction injury pain. Bottom: Overview of the timeline of systemic injection and behavioral metrics. Mice were injected intraperitoneally (i.p.) with either 10 mg/kg or 30 mg/kg KP daily after nerve constriction injury (PSNL). Pain behavior was measured 6 h post-injection. Bar diagram: Analgesic effects of KP for sensitized mechanical withdrawal were dose-dependent, namely less accentuated at 10 mg/kg and more pronounced at 30 mg/kg. Data are represented as mean values ± SEM. Dose–response was measured with simple linear regression, $p = 0.021$ (d4), $p = 0.0004$ (d7), $p < 0.0001$ (d11, 14). $n = 6$ for KP (10 mg/kg), $n = 7$ for KP (30 mg/kg) treatment, $n = 13$ for vehicle control; *$p = 0.0031$, ***$p = 0.0005$, $p = 0.0001$, ****$p < 0.0001$, mixed-effects statistics. **b** Systemic KP reduces mechanical allodynia in bone cancer pain evoked by infusion of mouse LLC lung cancer cells into the femur. Bottom: an overview of the timeline of systemic injection and metrics. Mice were injected with KP and tested as in (**a**). Bars: analgesic effects of KP for sensitized mechanical withdrawal were significant at d10 and 14 for 30 mg/kg vs. 10 mg/kg and vs. vehicle control. Data are represented as mean values ± SEM. $n = 5$ mice/group; *$p = 0.0012$, $p = 0.016$, **$p = 0.0057$, $p = 0.003$, two-way ANOVA. **c** Intrathecal (i.t.) KP reduces mechanical allodynia in neuropathic pain following nerve constriction injury. Bottom: Overview of the timeline of nerve constriction (CCI), i.t. injection and behavioral metrics. Bar diagram: analgesic effects of KP for mechanical allodynia were noted upon daily i.t. injection (30 μg), not in vehicle-injected controls. Withdrawal behavior was measured two times after each i.t. injection, +3, +24 h. Data are represented as mean values ± SEM. $n = 5$ mice/group, *$p = 0.04$, $p = 0.013$, **$p = 0.0027$, ****$p < 0.0001$, two-way ANOVA. **d** i.t. co-application of KP and specific KCC2 transport inhibitor, VU0240551 (30 μg), blocks the central analgesic effects of KP, which are apparent at +3 and +4 h post-i.t. Behavioral assays were conducted at 3, 4, and 5 h on day 7 after i.t. injection. Data are represented as mean values ± SEM. $n = 5$ mice/group; **$p = 0.0044$, ***$p = 0.0003$, two-way ANOVA. Source data are provided as source data files.

(Supplementary Fig. 3c). We also assessed whether KP (30 mg/kg; i.p.) can trigger brain reward mechanisms by measuring conditioned place preference (CPP)[46,47], and observed no such effects (Supplementary Fig. 3d). Thus, KP functions as an analgesic in relevant preclinical mouse models and does not cause sedation or unwanted effects on reward, coordination, and stamina.

**KP renormalizes $E_{\text{GABA}}$ in the SCDH by increasing *Kcc2* expression and function.** With the above findings and in view of the SCDH as the likely site of analgesic action of KP[3,6], we

investigated the effect of nerve injury on *Kcc2* expression and functioning in the superficial SCDH as well as its response to KP. We focused on these superficial layers, because laminae I–II play crucial roles in spinal cord nociceptive transmission[48], and *Kcc2* expression underlies normal functioning in laminae I–II neurons[3,6,49]. We remain aware that deeper laminae also play an active role in pain modulation. Projection neurons exist in all laminae and can contribute to the overall behavioral effect. We found that in the SCDH, KP treatment (10 mg/kg daily post-injury for 1 week) repaired attenuated *Kcc2* expression caused by PSNL nerve injury, at both the mRNA level (SCDH

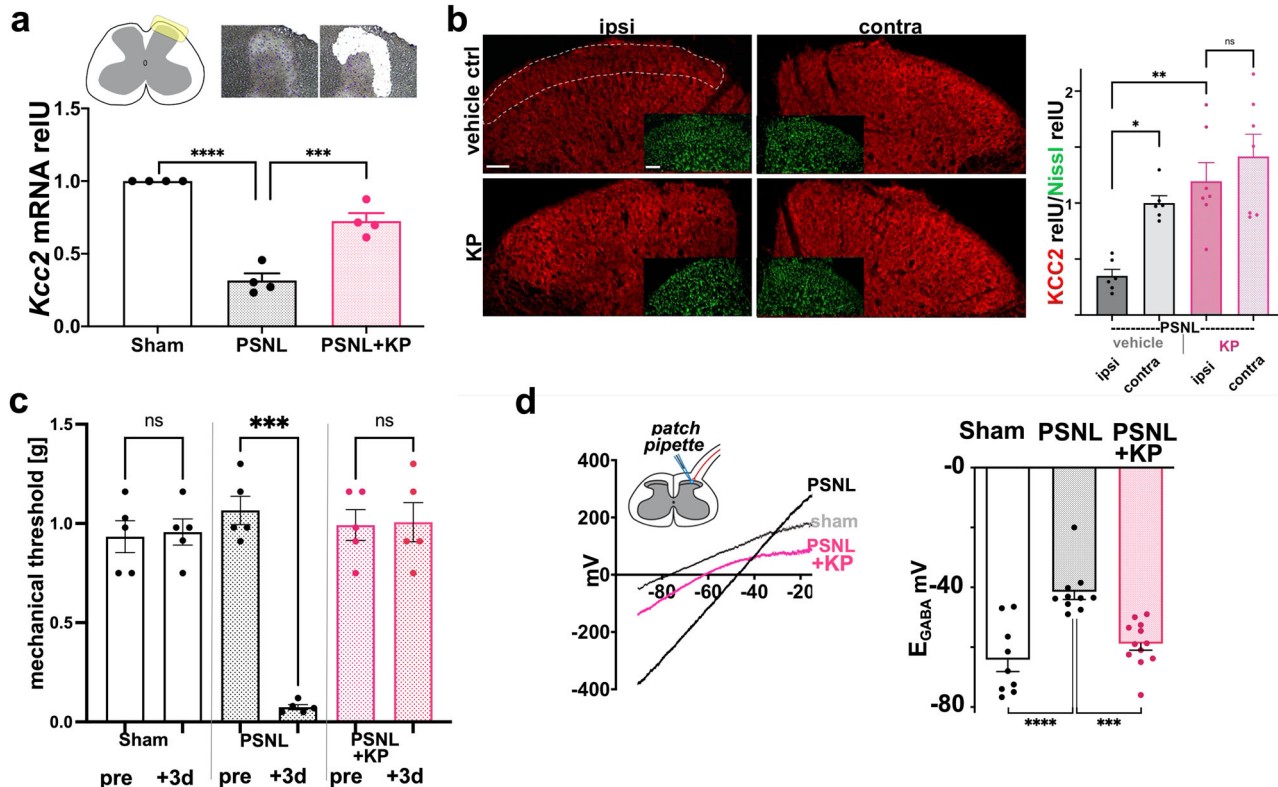

**Fig. 4 Kenpaullone re-normalizes $E_{GABA}$ in spinal cord dorsal horn by increasing *Kcc2* expression/KCC2 function in mice with neuropathic pain. a** Top left: Laminae I–II area of spinal cord dorsal horn (SCDH) is highlighted (yellow). Top right: image representation of laminae I–II area before and after laser capture microdissection. Bottom, bar diagram: KP (10 mg/kg) rescues significantly attenuated *Kcc2* mRNA expression in the SCDH after nerve constriction injury (PSNL) vs. vehicle-treated mice; KP injected daily for 7 days after PSNL, SCDH microdissected 16 h after last injection of KP. Data are represented as mean values ± SEM. $n = 4$ mice/group, ***$p = 0.0002$, ****$p < 0.0001$ one-way ANOVA. **b** KP treatment increases KCC2 expression in the SCDH. Left-hand micrographs: representative KCC2 immuno-staining ipsilateral vs. contralateral to injury (see antibody validation in Supplementary Fig. 2a) of the SCDH in nerve constriction injury (PSNL), region-of-interest for densitometric measurement of KCC2 in the SCDH outlined with dotted white line in upper-left micrograph, focus on Rexed layers I–II because of their relevance for neurotransmission of nociceptive afferent signals. KCC2 signals were normalized for Nissl stain (green fluorescent signal, inlet micrographs, see Supplementary Fig. 4a), which did not differ between conditions and sides, see Supplementary Fig. 4a, b. Scale bar in upper left micrograph and inlet = 100 μm, valid for all panels. Bar diagrams: KP (10 mg/kg; daily treatment for 7 days after PSNL) increases KCC2 protein expression in SCDH vs. vehicle control in PSNL nerve constriction, ipsilateral to injury, no difference in KCC2 signal contralaterally. Data are represented as mean values ± SEM. $n = 6$ mice/vehicle control group, $n = 7$ mice/KP group, *$p = 0.029$; **$p = 0.0026$, one-way ANOVA. **c** Mechanical hypersensitivity in juvenile mice (5–7 weeks of age) after PSNL (grey bars) and its complete behavioral recovery after early intervention with KP (30 mg/kg; pink bars; 1× injection pre-PSNL, every 24 h post-PSNL). Data are represented as mean values ± SEM. $n = 5$ mice/group for all groups, behavioral assessment 72 h after KP injection. Note accentuated responses for sensitization and rescue in younger mice when comparing with adult mice as shown in Fig. 3a. ***$p = 0.0005$ one-way ANOVA. **d** Perforated patch-clamp recordings showing Cl⁻-extrusion capacity ($E_{GABA}$) in lamina-II neurons from spinal cord slices of the same juvenile mice as treated in panel **c**, $E_{GABA}$ measured using the perforated patch method, illustrated by the schematic, 1–3 neurons per mouse. Left: representative I–V plot. Right: bar diagram showing quantification of $E_{GABA}$ indicating a significant depolarizing shift in sham vs. PSNL with vehicle treatment ("PSNL"). Note significant hyperpolarization in response to KP treatment in PSNL mice. Of note, PSNL plus KP was not different from sham injury. Data are represented as mean values ± SEM. $n = 9$ neurons (sham), $n = 10$ neurons (PSNL), $n = 12$ neurons (PSNL + KP); ****$p < 0.0001$, ***$p = 0.0004$, one-way ANOVA. Source data are provided as source data files.

microdissection, Fig. 4a), and protein level (morphometry after KCC2 immunolabeling with SCDH laminae I–II as region-of-interest, comparison ipsilateral vs. contralateral to injury, using Nissl stain for normalization of KCC2 signal; Fig. 4b, Supplementary Fig. 4a–c).

Nerve injury-induced downregulation of KCC2 has been shown to cause an increase toward more positive values of the chloride reversal potential in SCDH neurons after application of GABA ($E_{GABA}$) when using electrophysiologic measurements in spinal cord slices[3]. We, therefore, investigated whether repair of attenuated *Kcc2* expression after nerve injury was associated with the re-normalization of $E_{GABA}$. Prior to the interrogation of spinal cord slice preparations at 72 h post-injury, we demonstrated robust mechanical allodynia of young mice (5–7 weeks) by PSNL and its almost complete behavioral reversal by systemic early-

intervention treatment with KP (one-time treatment pre-PSNL, daily treatment post-PSNL for 3 days, 30 mg/kg) (Fig. 4c). Interestingly, PSNL nerve injury sensitized the juvenile mice more than adults. Also, behavioral rescue with KP was highly effective in juveniles. Next, in spinal cord slice preparations derived from these animals, SCDH lamina-II neurons were investigated by perforated patch-clamp for $E_{GABA}$ (Fig. 4d, Supplementary Fig. 4d, e). We focused on lamina-II neurons mindful of the functional organization of the SCDH in which noxious thermal and mechanical afferents are relayed in distinct laminae (but see above comment on co-contribution of deeper layers). Whereas thermal afferents are relayed in lamina-I and outer lamina-II, mechanical afferents are relayed predominantly in lamina-II[49], and our nocifensive behavioral measurements used mechanical cues throughout our study. Compared to sham animals, in which

a baseline $E_{GABA}$ of $-64.2 \pm 3.9$ mV ($n = 9$) was observed in lamina-II neurons, nerve injury resulted in a more excitable $E_{GABA}$ of $-41.9 \pm 2.6$ mV ($n = 10$) which was statistically significant. Strikingly, this robust $E_{GABA}$ increase in lamina-II neurons following nerve injury was reversed by KP treatment to yield an $E_{GABA}$ of $-58.8 \pm 2.2$ mV ($n = 12$) (Fig. 4d), which is consistent with the repair of *Kcc2* expression in the SCDH of KP-treated mice. Thus, in a constriction nerve injury model in mice, the central analgesic action of KP relies on the enhanced gene expression of *Kcc2* resulting in re-normalization of $E_{GABA}$ in pain-relaying neurons in the SCDH.

**Cellular mechanism of action in neurons: GSK3ß → δ-cat → Kaiso → *Kcc2*.** We next sought to determine whether KP's previously reported inhibitory actions on GSK3 and/or CDKs[37], known for two decades[25,26,50], was responsible for increasing expression of *Kcc2*. These studies were conducted in primary cortical neurons because: (1) these neurons were used for our initial screen, (2) there was rodent–human similarity in terms of the enhancing effect of KP on *Kcc2*/*KCC2* gene expression, and (3) the enhancing effects of KP on *Kcc2* gene expression and subsequently increased KCC2 chloride transporter function in primary cortical neurons were highly similar to our findings in SCDH lamina-II neurons, which cannot be cultured as readily for mechanistic cellular studies.

Using a set of known GSK3 inhibitors different from KP, we confirmed increased expression of *Kcc2* in vitro. In contrast, a suite of CDK-inhibitory compounds not only did not increase, but decreased expression of *Kcc2* (Fig. 5). Although KP inhibits both GSK3 and CDK, GSK3 is inhibited by at a far lower concentration ($EC_{50} = 25–75$ nM) than CDKs (several hundred nM)[25,37], suggesting KP's GSK3-inhibitory effects as the key mechanism of action by which it increases gene expression of *Kcc2*. Further support for this hypothesis was obtained by co-treating primary cortical neurons and N2a neuron-derived cells, which have moderate *Kcc2* expression and *Kcc2* gene-regulation similar to primary cortical neurons[18], with selective GSK3-inhibitor CHIR99201 (10 nM) and KP (0.5 μM). Compared to treatment with KP alone, we did not observe a significant increase of *Kcc2* mRNA abundance as a result of co-treatment (Supplementary Fig. 5a, b), suggesting no influence of the CDK-inhibitory effect of KP on *Kcc2* expression. Furthermore, aza-KP, a closely related analog of KP[37,51], which is a slightly more potent GSK3ß inhibitor than KP (18 vs. 25–75 nM) and 100-fold less active as a CDK inhibitor, significantly enhanced *Kcc2* mRNA in rat primary cortical neurons, whereas 20 nM KP did not (Supplementary Fig. 5c). While these results are in alignment with those of other tested GSK3ß inhibitors (see Fig. 5), aza-KP is chemically more similar to KP than any of the other inhibitors.

Collectively, the results led us to conclude that: (i) *Kcc2* gene expression-enhancing function of KP was observed with other GSK3-inhibitors, including the closely related structural analog of KP, aza-KP, (ii) *Kcc2* gene expression-enhancing effects were not observed with CDK-inhibitors, and (iii) KP-induced enhancement of *Kcc2* gene expression was not augmented when co-applied with the selective GSK3-inhibitor CHIR99201. Thus, it is conceivable that KP increases *Kcc2* gene expression solely via inhibition of GSK3, not CDK.

Direct binding of KP to GSK3ß (Fig. 6a, Supplementary Data File S2) in primary cortical neurons was confirmed using a drug affinity responsive target stability (DARTS) assay[52], which is consistent with previous studies in other cell lines[53], yet novel in primary neurons. Of note, binding to other known kinase targets

of KP, in particular CDKs, was not identified in our unbiased DARTS assay.

To further address the issue of KP/GSK3ß interaction, we then conducted molecular dynamics computational binding studies of KP to GSK3ß. Using this modeling, we found that KP binds to GSK3ß in the ATP binding pocket and that GSK3ß(V135) formed hydrogen bonds to the pyrroline-N of KP (Fig. 6a, Supplementary Fig. 6a). Importantly, KP-GSK3ß binding, as suggested by our molecular dynamics simulation mimicked faithfully a structurally verified binding of a selective GSK3ß inhibitor, compound-2, to GSK3ß[54]. Running compound-2 and GSK3ß through our molecular dynamics simulation replicated their binding as experimentally verified by X-ray crystallography. These findings extend our result with kinase inhibitors that selectively target GSK3 or CKDs, so that the likelihood increases that the GSK3ß-inhibitory function of KP is responsible for its *Kcc2*/*KCC2* gene expression-enhancing effect in neurons.

We next sought to identify kinase targets of GSK3ß that are differentially phosphorylated in response to KP. Using unbiased phosphoproteomics assays in rat primary cortical neurons, we identified the neuronal catenin, δ-catenin (CTNND2; δ-cat)[55,56]. We found that the serine at position 259 was one site at which differential phosphorylation occurred short-term (1 h) and persisted long-term (24 h) in response to KP (Fig. 6b, Supplementary Data File S3). The respective residue in human δ-cat is S276. ß-catenin (ß-cat) could also be a GSK3ß kinase target, yet we did not find ß-cat significantly differentially phosphorylated. δ-cat(S276) is found in phosphosite.org and has been previously described[57], but its identification in primary neurons is novel. Catenin phosphorylation facilitates its own intracellular degradation via ubiquitination[58,59]. To examine whether non-

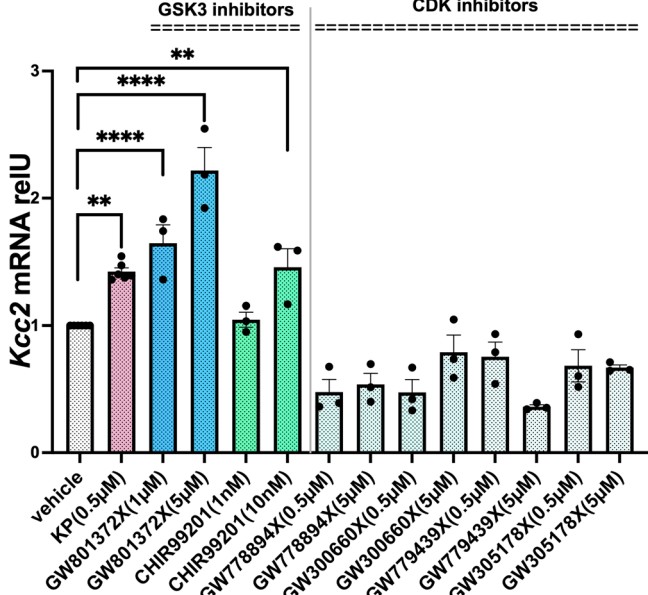

**Fig. 5 Kenpaullone and GSK3-inhibitors, not CDK-inhibitors, increase *Kcc2* expression in cultured central neurons.** GSK3-inhibitors increase *Kcc2* mRNA expression, measured by RT-qPCR, in a dose-dependent manner, whereas several CDK-inhibitors do not increase *Kcc2* mRNA expression, they rather reduce it. Rat primary cortical neurons treated with inhibitors at the indicated concentrations, treatment started at DIV5, harvested DIV8. Data are represented as mean values ± SEM of mRNA expression of 3–6 independent neuronal cultures, $n = 6$ for vehicle control and KP, $n = 3$ for all other treatments. **$p = 0.0018$, $p = 0.0075$, ****$p < 0.0001$, compound vs. vehicle, one-way ANOVA. Source data are provided as source data file.

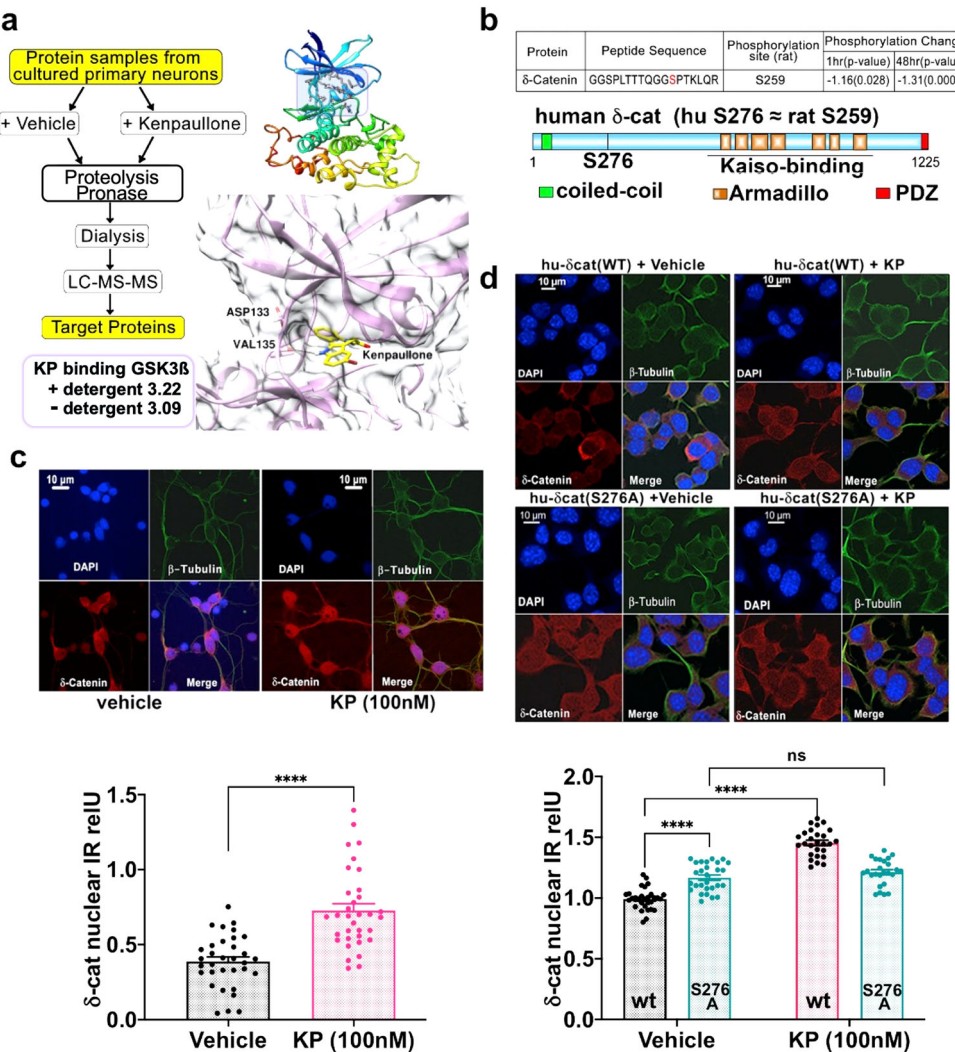

**Fig. 6 Cellular mechanism of action of Kenpaullone in central neurons. a** Left diagram: DARTS methodology to identify proteins that bind to KP in rat primary cortical neurons, KP applied for 30 h at DIV5 (see the "Methods" section/Supplementary Methods. KP binding to GSK3β is independent of detergent treatment of the protein sample preparation. Of note, KP binding to GSK3ß was documented whereas binding to CDKs was not (see Supplementary Data File S2 DARTS). Right-hand: upper structure is the crystal structure of human GSK3β (pdb 6V6L), the boxed area shows its ATP-binding pocket, note detail below with KP binding within the ATP pocket, with Val135 forming a hydrogen bond to the KP pyrroline-N, as determined by molecular dynamics simulation (see Supplementary Fig. 6a and Supplementary Information/Supplementary Methods). **b** Upper diagram: Phosphoproteomics assays reveal S259 phosphorylation target in δ-cat protein after KP treatment (1 μM for 1 h/24 h) of rat primary cortical neurons (DIV5), significant de-phosphorylation resulted after 1 h treatment and was sustained at 24 h (see Supplementary Data File S3 phosphoproteomics). Lower panel: schematic representation of the structure of human δ-cat (CTNND2) showing functional domains. Human residue S276 matches rat S259. The Armadillo domain region plays a key role in transcription factor Kaiso binding to δ-cat. **c** Top micrograph panel: Representative immuno-labeling of neuronal ßIII-tubulin (green) and δ-cat (red) before and after KP treatment (100 nM, DIV5–8) in rat primary cortical neurons, DAPI counterstain for nuclei (blue). Bottom bar diagram: KP significantly increases δ-cat nuclear abundance (relative abundance normalized for cytoplasmic abundance; see also Supplementary Fig. 6b and Supplementary Methods). Data are represented as mean values ± SEM. $n = 33$ (vehicle control), $n = 34$ (KP) neurons/ group,****$p < 0.001$ KP vs. vehicle, two-sided $t$-test. **d** Top panel: Representative immuno-labeling of ßIII-tubulin and δ-cat before and after KP treatment (100 nM for 3 days) in differentiated N2a mouse neural cells (see Supplementary Fig. 6e), which were transfected with either hu-δ-cat(WT) or hu-δ-cat(S276A). Bottom Panel: KP significantly enhances the nuclear abundance of δ-cat when transfected with δ-cat(WT); same measurement method as in (**c**), mean of hu-δ-cat(WT)/vehicle-treated subgroup set as "1" with a subsequent adaptation of all other groups. Mutation hu-δ-cat(S276A) increases nuclear abundance, but treatment with KP has no significant effect on cells expressing it. Data are represented as mean values ± SEM. $n = 25$–30 neurons/ group ($n = 30$ WT/vehicle control, $n = 27$ WT/KP; $n = 28$ S276A/vehicle control, $n = 25$ S276A/KP) ****$p < 0.0001$ vs. vehicle, mixed effects statistics. Source data are provided as source data file.

phosphorylated δ-cat shows increased nuclear abundance as suggestive evidence of its trafficking to the neuronal nucleus, we conducted specific δ-cat immunolabeling followed by confocal microscopy and morphometry, neuronal cultures treated with KP. Morphometry shows that KP treatment led to increased δ-cat nuclear abundance in rat primary cortical neurons. δ-cat nuclear abundance was determined relative to its abundance in the cytoplasm (Supplementary Fig. 6b, Supplementary Methods), thus validating its nuclear increase (Fig. 6c). Mindful of the known binding of ß-cat/δ-cat[60], we immunolabeled for ß-cat and obtained a similar result (Supplementary Fig. 6c, d), which is congruent with our δ-cat nuclear abundance findings.

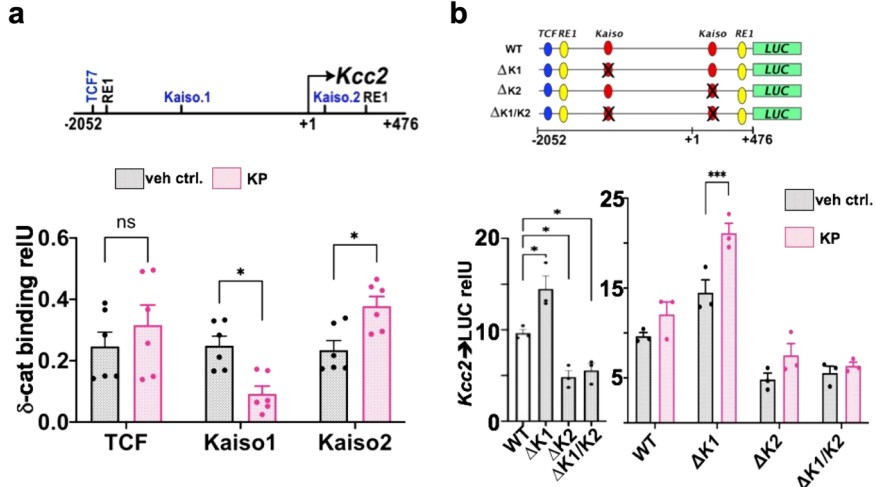

**Fig. 7 Delta-catenin binds to the proximal *Kcc2* promoter at two Kaiso sites, regulated by Kenpaullone. a** Structure of mouse *Kcc2* gene encompassing 2.5 kb surrounding the transcription start site (TSS; +1). Location of DNA-binding sites: Kaiso1 (−1456 to −1449), Kaiso2 (+83 to +90) and TCF (−1845 to −1838) relative to TSS, all three sites can bind δ-cat via Kaiso (Kaiso1, −2 sites) and ß-cat (TCF). Bottom, bar diagram: Chromatin immuno-precipitation (ChIP) using anti-δ-cat antibody in rat primary cortical neurons reveals binding of δ-cat to all three sites. KP treatment (100 nM, DIV5-8) significantly increases binding of δ-cat to the *Kcc2* promoter on the Kaiso2-binding site, significantly reduces binding to Kaiso1, and no significant increase at TCF. Data are represented as mean values ± SEM. $n = 6$ independent neuronal cultures were subjected to ChIP; *$p = 0.011$, $p = 0.018$; KP-treatment vs. vehicle, two-sided *t*-test. **b** Top panel: mouse *Kcc2* promoter constructs, Kaiso1, −2 were deleted also a ΔK1/K2 construct was built devoid of both sites. Dual-RE1 sites and TCF site is shown for orientation, also TSS at +1. Bottom, bar diagrams: Luciferase (LUC) activity of *Kcc2* promoter constructs in N2a cells with neuronal differentiation (Supplementary Fig. 6a), treatment with 100 nM KP for 3 days. Left-hand: ΔK1 significantly increased over WT, ΔK2 and ΔK1/K2 significantly decreased vs. WT, $n = 3$ independent culture per transfection, *$p = 0.014$, $p = 0.014$, $p = 0.032$, 1-way ANOVA. Right-hand: repressive Kaiso1-binding site deletion with enhanced activity of the *Kcc2* promoter, significantly enhanced further in response to KP. Data are represented as mean values ± SEM. $n = 3$ independent cultures per transfection; ***$p = 0.0003$ KP-treatment vs vehicle for the respective construct, two-sided *t*-test. Source data are provided as source data files.

To investigate the likelihood that S276 is a relevant phosphorylation site in δ-cat and a GSK3ß kinase target in neurons, we again used N2a neural cells because these cells transfect at higher efficiency than primary cortical neurons. In our cultures, N2a cells expressed neuronal ßᵢᵢᵢ–tubulin in elongated processes (Fig. 6d, Supplementary Fig 6e) as an indicator of their neuronal differentiation, which was more pronounced in response to 0.2 µM KP. Furthermore, when transfected with human δ-cat(WT), KP significantly enhanced nuclear abundance of δ-cat in these cells (Fig. 6d), which was very similar to findings recorded in primary cortical neurons, therefore again validating the N2a cell line. To further characterize the function of S276 of δ-cat as a GSK3ß kinase phosphorylation target, we then transfected neuronalized N2a cells with δ-cat(S276A), a phosphorylation-resistant mutant of δ-cat(S276), which resulted in significantly increased nuclear abundance of δ-cat(S276A) that was not increased further upon KP treatment (Fig. 6d).

Thus, δ-cat(S259/S276) (rat/human) is very likely a relevant phosphorylation site in δ-cat and a GSK3ß kinase target in neurons, which is supported by our findings that inhibition of GSK3ß or rendering S276 phosphorylation-resistant enhances the nuclear abundance of δ-cat, as well as a nuclear abundance of its known binding partner ß-cat.

We conclude that KP enhances *Kcc2* gene expression in neurons, while at the same time increasing the nuclear abundance of catenins. This raises the exciting possibility that catenins regulate *Kcc2* gene expression at the *Kcc2* promoter. Consequently, we explored catenin effects on the *Kcc2* promoter.

We identified two Kaiso binding sites (known binding sites for δ-cat[29,61]) in the *Kcc2* proximal promoter using computational methods[62] which bracketed the transcriptional start site (TSS) of the *Kcc2* gene (Fig. 7a). In rat primary cortical neurons, δ-cat was bound to both Kaiso sites in the *Kcc2* promoter (Fig. 7a). The

binding of δ-cat to the site 3′ to the TSS was enhanced in response to KP treatment, whereas binding to the upstream 5′ site was inhibited, suggesting a complex regulation in that the 5′ site functions as a repressor and the 3′ site as enhancer. ß-cat bound to a T-cell factor (TCF) DNA-binding site[63] close to the 5′ RE-1 site within the *Kcc2* promoter (Supplementary Fig. 7a)[18], and treatment of cells with KP significantly increased this interaction, indicative of enhancement. Congruent with these DNA-binding studies, treatment of primary cortical neurons with small molecule catenin-inhibitor ICG caused significant decrease of *Kcc2* mRNA abundance (Supplementary Fig. 7b).

To validate the relevance of the Kaiso- and TCF-binding sites, we built promoter expression constructs with rationally targeted deletions to interrogate the effects of these sites on the activity of the *Kcc2* promoter (Fig. 7b). These molecular tools would also allow us to determine if the activity of the Kaiso- and TCF-sites was regulated by KP. For ease of transfection, key to this method, we again used N2a neural cells. We found that the 5′ and 3′ δ-cat Kaiso-binding sites functioned in repressive and enhancing manners, respectively (Fig. 7b). The presence of the 3′ δ-cat Kaiso binding site and absence of the 5′ site led to significantly enhanced activity of the *Kcc2* promoter upon treatment with KP. Deletion of both Kaiso sites led to markedly reduced promoter activity and non-responsiveness of the construct to KP. Deletion of the TCF binding site from the *Kcc2* promoter did not change *Kcc2* promoter activity or its response to KP treatment (Supplementary Fig. 7c). The triple-deletion of both Kaiso sites and the TCF site rendered the construct minimally active and completely non-responsive to KP.

These data strongly suggest that δ-cat, a kinase target of GSK3ß in CNS neurons as we demonstrate, shows enhanced nuclear abundance when inhibiting GSK3ß with KP. In the nucleus, δ-cat interacts with and binds to the *Kcc2* promoter to enhance *Kcc2*

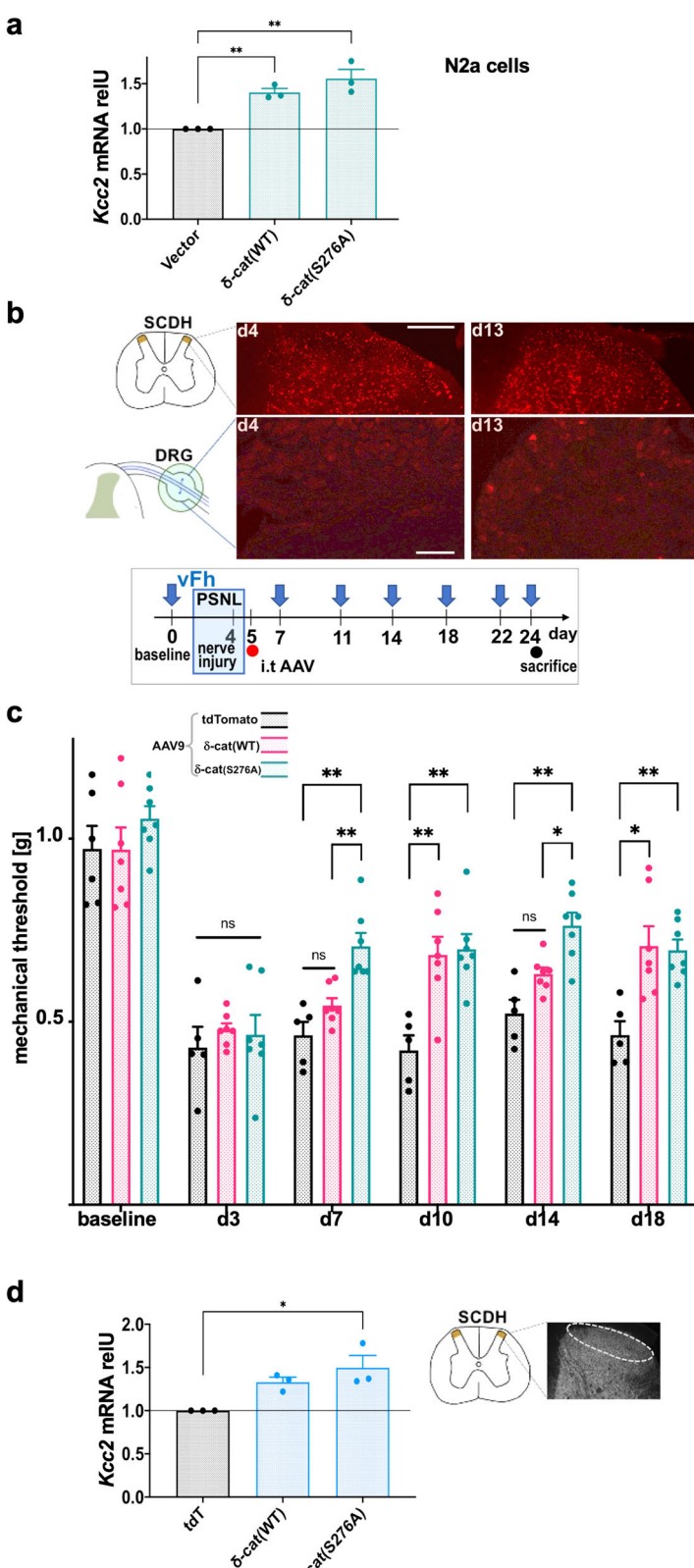

expression via two Kaiso DNA-binding sites. ß-cat, as a known nuclear binding partner to δ-cat which binds to a TCF site in the *Kcc2* promoter, also shows increasing nuclear abundance and binding in response to KP. ß-cat however, is not a significant neuronal GSK3ß kinase target (Supplementary Data File S3), and likely plays a supportive-ancillary role in the enhancement of *Kcc2* gene expression.

**δ-cat overexpression in the spinal cord inhibits neuropathic pain after nerve constriction injury**. We hypothesized that δ-cat when expressed as a transgene in spinal pain relay neurons, will facilitate analgesia in nerve constriction injury. We first documented that a δ-cat transgene increases *Kcc2* expression in N2a neural cells, and that *Kcc2* expression levels were slightly elevated when using δ-cat(S276A) (Fig. 8a). Thus, human δ-cat

**Fig. 8 δ-cat spinal transgenesis is analgesic in nerve constriction injury. a** δ-cat(WT) transgene significantly increases *Kcc2* mRNA expression in differentiated N2a cells (48 h post-transfection), and expression level was slightly elevated when transfecting δ-cat(S276A), both significantly increased over WT. Data are represented as mean values ± SEM. *n* = 3 independent cultures were subjected to transfection with the δ-cat constructs; **p* = 0.0014, *p* = 0.0075, δ-cat construct vs. control transfection, one-way ANOVA. **b** Top, schematics and micrographs: SCDH lamina I–II showing appreciable early (d4) and longer-term (d13) td-Tomato expression driven by hu-synapsin promoter after i.t. injection of AAV9-tdTomato; note sparse staining in L5 DRG at these time-points. Scale bar upper panel = 200 μm, lower panel = 100 μm. Micrographs are representative for 4 micrographs derived from 2 mice/time-point, of spinal segment L5 and L5 DRG. Lower schematic: Timeline for behavioral testing after constriction nerve injury and subsequent i.t. injection of AAV9 for directed expression of δ-cat(WT), δ-cat(S276A) and tdTomato. **c** Mechanical withdrawal thresholds after nerve constriction injury (PSNL). Significant improvement of sensitization for both δ-cat(WT) and δ-cat(S276A) constructs with sustained benefit over 2 weeks. Note more potent analgesia by δ-cat(S276A) based on the data on d7, d14 showing effectiveness for δ-cat(S276A), not for δ-cat(WT). Data are represented as mean values ± SEM. *n* = 6 mice/tdTomato ctrl, *n* = 7 mice/δ-cat transgenes; **p* < 0.05, ***p* < 0.01, d7: *p* = 0.0024, *p* = 0.0083; d11: *p* = 0.0058, *p* = 0.0022; d14: *p* = 0.0027, *p* = 0.021; d18: *p* = 0.011, *p* = 0.0029; mixed effects statistics. **d** *Kcc2* mRNA in microdissected SCDH (d7) was significantly increased in δ-cat(S276A) as assessed by RT-qPCR. For δ-cat(WT), *Kcc2* abundance was elevated over tdTomato, but not to significant levels. Data are represented as mean values ± SEM. *n* = 3 mice/group, **p* = 0.013 for δ-cat(S276A) vs. tdTomato control, one-way ANOVA. Source data are provided as source data file.

transfection is sufficient to mimic the effects of KP in a mouse neural cell line.

We then constructed AAV9 vectors harboring human δ-cat and δ-cat(S276A), driven by the minimal human neuronal synapsin promoter (huSyn[64]), known to drive expression in neurons, not glia. We used AAV9 and huSyn because in a previous in-depth study, AAV9 harboring fluorescent reporter driven by huSyn, upon spinal injection, readily transduced spinal neurons and spared DRG primary afferent neurons[65]. However, whether this was also the case for i.t. injection needed to be examined.

We used synapsin-tdTomato as a control fluorescent reporter and injected $5 \times 10^9$ viral genomes (5 μL; i.t.) of each construct. Assessment of tdTomato fluorescence on d4 after injection revealed spinal transgene expression that was evenly manifested in the SCDH (Fig. 8b). This pattern and abundance of tdTomato + cells did not change when examining at d13. We also looked for tdTomato-expressing cells in lumbar DRGs, both on d4 and d13, yet found them only sparsely for both time-points. These findings point to a predominant expression of the transgenes in the spinal cord, including SCDH, and rather sparse and protracted expression in peripheral DRG sensory neurons.

When viewing our present findings vs. the above-mentioned previous study[65], it appears that our i.t. injection of AAV9, with the viral transgene driven by the minimal human synapsin promoter, evokes robust and earlier transgene expression in the spinal cord gray matter including SCDH, whereas expression in the DRG remains relatively sparse and protracted. Thus, even though intraspinal injection as practiced by Haenraets et al. [65] is fundamentally different from i.t. delivery as practiced here, our pattern of fluorescent reporter gene expression, robust in the spinal cord gray matter including SCDH, and sparse in the DRG, was related to that previously reported[65]. We then measured mechanical withdrawal thresholds after nerve constriction injury (PSNL) (Fig. 8b, c) and found that δ-cat(S276A) expression led to significantly reduced mechanical allodynia, of note with a delayed start of effect beginning on d7 post-injection, and sustained duration of action until the end of experiment at d18 (Fig. 8c). In this study, δ-cat(S276A) showed a statistically significant increase in effect vs. δ-cat(WT) on d7 and d14. Compared to controls, δ-cat(WT) was significantly effective on d11 and d18. Congruent with behavioral findings, *Kcc2* mRNA in microdissected SCDH was significantly increased in δ-cat(S276A) (Fig. 8d). For δ-cat(WT), *Kcc2* mRNA abundance was higher than in controls but the difference was not statistically significant. A caveat to interpretation of this result is that only a fraction of sensory relay neurons in the total measured pool of neurons in the SCDH were virally transduced (Fig. 8b). Therefore, a pool of non-transduced SCDH neurons reduced the total fractional amount of

measured *Kcc2* mRNA produced by neurons expressing δ-cat transgenes, conceivably below the level required to demonstrate a significant difference from control. Regardless, we have demonstrated that δ-cat(S276A) spinal transgenesis via i.t.-injected AAV9 is sufficient to evoke analgesia after nerve constriction injury, an effect also seen, but to a lesser degree, with δ-cat(WT). Importantly, δ-cat(S276A) spinal transgenesis was accompanied by increased expression of *Kcc2* in the SCDH. These in-vivo findings in a neuropathic pain model lend support to our postulated mechanism of *Kcc2* gene expression-enhancement by KP, specifically via nuclear action of δ-cat at the *Kcc2* promoter, as we established in primary neurons and neural cells. Moreover, our discovery points toward a genetically encoded approach that can be developed for translation into clinical use as an alternative or complement to KP-related small molecules.

## Discussion

Accumulating evidence suggests that injury-induced reduction in expression and function of the neuronal chloride extruding transporter, KCC2, in the dorsal spinal cord critically contributes to chronic pathologic pain[3,6,48]. This concept led us to search for a small molecule which can increase *Kcc2* gene expression. Using measures of *Kcc2* promoter activity, *Kcc2* mRNA abundance, and [Cl$^-$]i, we conducted an unbiased screen in primary cortical neurons, taking advantage of two NCI libraries containing 1057 compounds that inhibit the growth of malignantly transformed cells. We reasoned that a sizable number of these compounds would interfere with the epigenetic and gene expression machinery of cancer cells, which renders them suitable candidates to function as *Kcc2*/*KCC2* expression-enhancers in non-dividing neurons.

Our screen identified KP, a GSK3/CDK kinase inhibitor with additional, advantageous neuroprotective properties[25,26,33–36]. Our studies of KP revealed the following principal findings:

1. KP directly enhances *Kcc2*/*KCC2* gene expression, not KCC2 transporter function, in a concentration-dependent manner and lowers [Cl$^-$]i in cultured mouse, rat, and human neurons;
2. Systemic administration of KP to mice attenuates neuropathic pain and bone cancer pain in a dose-dependent manner;
3. Intrathecal administration of KP to mice attenuates neuropathic pain depending on spinal KCC2 chloride transporter activity;
4. Systemic administration of KP to mice with constriction nerve injury repairs defective *Kcc2* gene expression in SCDH neurons and re-normalizes their GABA-evoked chloride reversal potential to more negative and electrically stable measures;

5. The mechanism by which KP enhances *Kcc2* gene expression likely involves binding to and inhibiting GSK3ß. This in turn inhibits phosphorylation of δ-cat at position S259 in rats (S276 in humans), which increases nuclear abundance of δ-cat which subsequently binds to and enhances the *Kcc2* promoter via two Kaiso-binding sites surrounding the TSS of the *Kcc2* gene.

6. Examining whether this mechanism could contribute to analgesia in live animals, we observed that spinal transgenesis of δ-cat(S276A) was sufficient to reverse neuropathic pain and to repair attenuated *Kcc2* mRNA expression in the SCDH.

Our analgesic approach as described herein is innovative. In the context of lowering neuronal chloride in the SCDH for analgesia, the previous discovery of small molecules that enhance KCC2 chloride extrusion is noteworthy[6]. This latter study brought full circle the initial landmark observation of lack of expression of KCC2 in SCDH neurons after peripheral nerve injury[3]. Together, these previous studies demonstrate the causality of KCC2 downregulation in pathologic pain which could possibly extend to human spinal pathophysiology in pathologic pain[22].

We conclude that KP and the identified GSK3ß → δ-cat → Kaiso → *Kcc2* signaling and gene-regulatory pathway may represent a strategic bridgehead for therapeutics development for the treatment of pathologic pain. Beyond pain, this strategy could also apply to other neurologic and mental health conditions in which restoration of KCC2 function is important, such as epilepsy, spinal cord and brain traumatic injury, neurodegeneration, and neurodevelopmental disorders. Our proposed analgesic mechanism is summarized in Fig. 9.

Though KP is predicted to have multiple targets in CNS neurons, we provide evidence that KP functions by binding to neuronal GSK3ß, but not to CDKs. We remain aware that KP can inhibit other kinases, but our data suggest that inhibition of GSK3ß and subsequent enhancement of *Kcc2* gene expression via δ-cat are important, perhaps dominant mechanisms of analgesic action of KP and that an important site of action is SCDH pain-relaying neurons. As it is known that KP inhibits GSK3ß with the highest potency from amongst known targets[25,51,53], we present further evidence that GSK3ß inhibition by KP might directly upregulate *Kcc2* gene expression via the δ-cat–Kaiso pathway. Neither this mechanism of a GSK3-inhibitory compound nor its conserved effect in human primary neurons has been reported previously. Additionally, while δ-cat-Kaiso almost certainly affects multiple neuronal genes, our data in SCDH neurons suggest that enhanced *Kcc2* gene expression and KCC2 function are the major analgesic effector mechanisms of KP. In addition, while δ-cat very likely has multiple target genes in SCDH neurons, our data suggest that spinally expressed δ-cat transgenes function as *Kcc2* gene expression-enhancers. Moreover, in neural cells, increased expression of δ-cat evokes increased *Kcc2* expression, whereas inhibition of catenins attenuates *Kcc2* expression. In animal models of neuropathic pain, our δ-cat spinal transgenesis leads to delayed onset but prolonged duration of analgesia, which is indicative of gene regulation and similar to the time profile of analgesia following KP treatment. Furthermore, SCDH *Kcc2* mRNA expression is significantly elevated for the δ-cat(S276A) transgene. Future studies will seek to determine the possible co-contribution of other δ-cat target genes in SCDH neurons.

While less supported mechanistically but translationally more relevant, another argument in support of our *Kcc2* expression-enhancing strategy is the absence of unwanted effects on choice behavior, motor stamina, and coordination. Effective targeting of

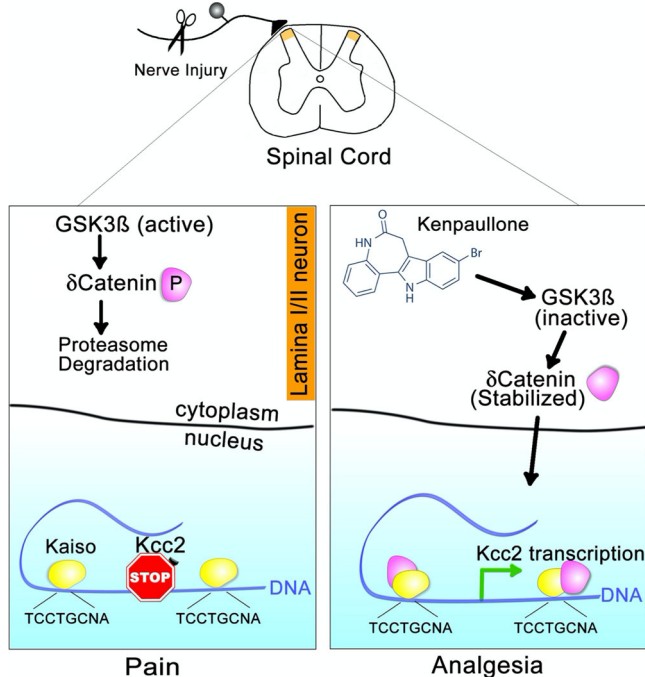

**Fig. 9 Postulated analgesic mechanism of action of Kenpaullone.** Left panel: Nerve injury facilitates activation of GSK3β kinase in pain relay neurons in layers I–II of the SCDH. GSK3ß phosphorylates δ-cat in the cytoplasm at S259(rat)/S276(human). Phospho-δ-cat is unstable, undergoes ubiquitination and subsequent degradation. In the nucleus, Kaiso recruits repressive transcription factors on the *Kcc2* gene promoter, leading to an overall repressed *Kcc2* transcription. Right panel: Treatment with KP inactivates GSK3β in SCDH layers I–II pain relay neurons. In turn, this leads to dephosphorylation and stabilization of δ-cat which enters the nucleus. In the nucleus, δ-cat binds to the Kaiso transcription factor complex and displaces repressive transcription factors. This leads to a net enhancement of *Kcc2* expression.

multiple pathways would likely negatively impact these behaviors. Moreover, the behavioral profile for *Kcc2* gene expression-enhancing KP was similarly benign as that of KCC2 chloride-extrusion enhancing compounds[6].

Our screening strategy and subsequent validation allowed us to address a fundamental problem of pathologic pain. This vastly unmet medical need, rooted in its chronicity and overall debilitating impact, is driven by genetic reprogramming, which results in a maladaptive phenotype[5,9,12,43,66]. One very important mechanism to contribute to the maladaptive phenotype is attenuated expression of *Kcc2* because of its impact on inhibitory transmission in pain-mediating neural circuits[3,6,7,20,21,23]. This has been postulated to contribute to the pathogenesis of other CNS disorders[67,68]. Regulation of *Kcc2* gene expression by GSK3ß and its kinase target δ-cat is a novel finding of our study. This insight will permit rational exploration of links between GSK3ß → δ-cat and attenuated *Kcc2/KCC2* gene expression and the resulting malfunction of inhibitory neurotransmission in chronic pain and several other relevant neurologic and psychiatric conditions, such as epilepsy, traumatic brain/spinal cord injury, Rett Syndrome, autism spectrum disorders, and perhaps Alzheimer's disease and other neurodegenerative diseases[69–77]. We selected KP because of its previously reported neuroprotective properties for spinal motoneurons, brainstem auditory relay neurons, and hypoxia-injured hippocampal neurons[33–36,78,79]. It is possible that there was a non-appreciated unifying mechanism

of *Kcc2* expression-enhancement by KP in these previous studies. Neuroprotective properties for a novel analgesic are welcome because chronic pain is associated with non-resolving neural injury mediated by neuroinflammation[80].

In a recently published elegant and pioneering study, small molecules were selected to enhance the expression of *Kcc2/KCC2* for the effective rescue of a modeled Rett Syndrome phenotype[81]. In this study, genome-edited human reporter neurons were screened against almost 1000 compounds, yielding enhancement of *KCC2* expression by compounds with four different pharmacological properties, namely kinase inhibition selective for FLT3 and GSK3ß, and activation of sirtuin1 and TRPV1 signaling. Kinase inhibitors (BIO—targeting GSK3ß; KW-2449—targeting FLT3) were used to effectively treat a preclinical Rett Syndrome model[77,81,82]. This study strengthens our results with KP as we suggest that KP's analgesic properties are primarily rooted in its GKS3ß-inhibitory effects.

Our finding that KP acts as an analgesic in a bone cancer pain preclinical model is also noteworthy with regard to cancer pain in general. This finding requires additional research to further elucidate the underlying cellular and neural circuit mechanisms, e.g. in the SCDH. Interestingly, effective analgesia was observed at the higher KP dose whereas bone lesions caused by implanted cancer cells were not significantly affected. However, given the important established roles of CDKs and GSK3 in cancer cells' growth, future studies will elucidate whether: (i) KP can enhance co-applied cell-ablative strategies, (ii) higher doses of KP might improve analgesia and function antineoplastically, (iii) attractive co-medications with complementary mechanisms-of-action, such as HDAC-inhibitors, can enhance the antineoplastic as well as analgesic potencies of KP.

Currently, there are no ongoing trials listed in clinicaltrials.gov seeking to repurpose GSK3ß inhibitors into analgesics. Ongoing clinical development of GSK3ß-inhibitory tideglusib, which is in phase-II trials for congenital myotonic dystrophy (NCT02858908)[83], could conceivably lead to its repurposing for pain, following approval for its primary, proposed indication. Even if not imminent anytime soon, since safety data appear to be re-assuring, a clinical trial for pathologic pain with tideglusib as a clinically well-developed GSK3ß inhibitor can now be envisioned.

It is our view that enhanced *Kcc2/KCC2* gene expression, based on KP treatment or δ-cat transgenesis as presented here, will complement direct enhancement of KCC2 chloride extrusion in targeting pathologic pain. Complementary use will help overcome recalcitrant lack of expression and function of KCC2 in pain relay neurons, as might be expected in clinical cases of "refractory" chronic pain. Clinical combination-use of *KCC2* expression-enhancers with analgesic compounds that have complementary mechanisms of action will be advantageous because the re-normalized inhibitory transmission will likely improve the efficacy of other compounds, such as gabapentinoids. Supporting this concept, a very recent study reported effective analgesia of GABA$_A$ receptor α2/α3 agonists in neuropathic pain in the presence of enhanced KCC2 chloride efflux[84].

## Methods

**Screening in primary cortical neurons from *Kcc2*-LUC transgenic mice, rat primary cortical neurons**. Transgenic mice that express red-shifted LUC under the control of the *Kcc2* promoter (−2052/+476, described in ref. [18]), inserted into the Rosa26 locus, were described by us in refs. [30,31]. We generated primary cortical neuronal cultures from newborn (p0) mice of this line[30,31].

After one week in culture, neurons were treated with compounds at 100 nM for 48 h; LUC activity was determined for each compound with vehicle and trichostatin-A as negative and positive controls, results shown in Supplementary Fig. 1a. Experimental flow is depicted in Fig. 1.

Rat primary cortical neuronal cultures were maintained as in ref. [18].

**Compound libraries**. NCI compound libraries Natural Products II and Mechanistic Diversity Set II were obtained from NCI, containing 1057 compounds (Supplementary Data File S1). See also https://dtp.cancer.gov/organization/dscb/obtaining/available_plates.htm.

**Human neuronal cultures**. The neuron-enriched cultures were established from male and female human fetal cortical specimens at 15–20 weeks of gestation. The protocols for tissue processing complied with all US federal and institutional guidelines at UC Irvine and Duke University. Human cortical neural cultures were set up and maintained according to refs. [30,85]. At DIV6, cultures were exposed to KP (50, 100, 400 nM) for 2–4 days and then harvested for isolation of RNA. For immunostaining, the experiment was repeated in independent cultures using 400 nM KP. Cells were fixed with 4% paraformaldehyde and processed for immunolabeling for KCC2, and neuronal maturation marker, synaptophysin, and then inspected and image-captured on a fluorescent confocal microscope (Zeiss LSM700; Imaris software for quantification of signal).

**δ-cat DNA constructs, transgenesis vectors**. Human δ-cat isoforms (δ-cat(WT), and δ-cat(S276A) point mutation) were cloned into pAAV-hSyn, tdTomato was used as control. AAV9 particles were packaged (Duke Viral Vector Core Lab). For i.t. injections, a dilution of $10^{12}$ viral genomes/mL was used.

***Kcc2* promoter-LUC reporter assays**. After identifying TCF and Kaiso-binding sites within the *Kcc2b* regulatory region (position −2052 bp to +476 bp relative to the TSS), these specific sites were deleted by PCR-based cloning. Following previous methods[18], the WT promoter and the respective engineered mutant promoters were cloned into pGL4.17 to drive LUC. N2a cells were transfected with these constructs, co-transfected with Renilla LUC for normalization, and assayed for dual-LUC activity 24 h after transfection. Mutant promoters were compared to WT, and the response to KP (500 nM) by the respective promoter was measured.

**Chemicals**. KP compound was synthesized by the Duke Small Molecule Synthesis Facility to >98% purity, verified by LC/MS. CHIR99201 and VU0240551 were obtained from Tocris. GW801372X, GW778894X, GW300660X, GW779439X, and GW305178X were supplied by the Structural Genomics Consortium (SGC) at UNC-Chapel Hill.

**Animals**. C57BL/6J male mice (10–12 weeks old; 4 weeks old) were obtained from The Jackson Lab (Bar Harbor, ME). *Kcc2*-LUC mice were generated by the Liedtke-Lab at Duke University and continued internally. All animal procedures were approved by the Duke University IACUC and carried out in accordance with the NIH's Guide for the Care and Use of Laboratory Animals. Mice were housed in a temperature- and moisture-controlled environment, 12/12 h light–dark cycle with laboratory mouse certified food available ad libitum as well as water.

**RT-qPCR from cultured neuronal cells and microdissected SCDH**. oligodT-initiated reverse transcription of 1 μg total RNA, DNA-se treated, was subjected to RT-qPCR using primers specific for *Kcc2* for rat and KCC2 for human sequences, normalized for neuronal β$_{III}$-tubulin, as described in ref. [18], primers as listed in Supplementary Table. For cultured neurons, total RNA was extracted from pelleted cells, and for spinal cord tissue, it was extracted from the microdissected lumbar SCDH.

**Behavioral assessments**. For pain-related assays, hind paw withdrawal in response to mechanical stimuli was assessed with von Frey hairs (vFH) using an automated vFH apparatus (Fig. 3a), or with vFH with logarithmically increasing stiffness (0.02–2.56g, Stoelting), and the 50% paw withdrawal threshold was calculated (Fig. 3b-d) [86,87]. Sensitization was implemented by sciatic nerve injury via ligation, using the well-established neuropathic pain models, PSNL and CCI[38,39]. For in vivo behavioral assays, mice were injected starting on day 1 after constriction injury with 10 or 30 mg/kg KP, intraperitoneally (i.p.), or i.t. at 30 μg in 5 μL.

Intrathecal compound or viral vector injection was conducted as in ref. [88].

Vestibulomotor function, cerebellar coordination, and motor stamina were assessed using an automated rotarod (RR) (Ugo Basile, Italy)[45]. Animals were RR-trained pre-injection of KP, which was injected daily, and their stamina/RR performance was recorded for 2 weeks.

CPP was conducted as in ref. [89] by recording choice behavior for one of two accessible chambers after 7 days of conditioning; animals were treated with KP at 30 mg/kg or vehicle in one chamber. CPP scores were calculated as post-conditioning time minus preconditioning time spent in the treatment-paired chamber.

**Bone cancer pain model**. Murine lung carcinoma cell line LLC1 (ATCC CRL-1642) was injected into mouse femora following previous protocol[41]. Under brief general anesthesia, $2 \times 10^5$ tumor cells were injected into the left femur's cavity. Mechanical sensitivity was assessed as described above for nerve constriction

injury. The osteolytic bone lesion was determined by faxitron, radiographs read by blinded investigators as described in refs. [40,41].

**Chromatin immunoprecipitation.** ChIP assay was carried out as described in ref. [18] using primary cortical neurons ($0.7 \times 10^6$).

**Cultured neuron immuno-cytochemistry.** Immuno-cytochemistry labeling of cultured neuronal cells was carried out as described in ref. [18] using antibodies specific for δ-cat, ß-cat, ß$_{III}$-tubulin, and FLAG epitope (see Supplementary Table). Imaging and morphometry of acquired images were conducted using a Zeiss LSM710 with Zen software, ImageJ for morphometry.

**Cultured neuron KCC2 protein abundance assay—virtual Western blot.** Primary rat cortical neuron cultures were treated with 0.5 μM KP DIV5–8 for immunodetection of size-separated protein extract for KCC2, as in refs. [90,91]. On DIV8, neuronal cultures were lysed and protein extract size-separated in a WES capillary electrophoresis unit (ProteinSimple, San Jose, CA, USA), immobilized through the photoactivation-capture method, and immunodetected using primary N-terminal KCC2 antibody[92,93], and secondary detection with HRP-coupled antibody and chemiluminescence. The chemoluminescent signal was measured (as in refs. [90,91]) and represented as a virtual blot.

**Spinal cord immuno-histochemistry.** Immunohistochemistry of the lumbar spinal cord was conducted as in refs. [31,94,95] using KCC2-specific antibodies. Nissl stain of spinal neurons (ThermoFisher/Invitrogen cat# N21480) was used for normalization of the KCC2 quantification, as in ref. [96].

**Chloride imaging.** Chloride imaging of primary cultured cortical neurons was conducted as in refs. [18,31,32]. Clomeleon ratiometric fluorescent chloride indicator protein was transfected into neurons and ratios were acquired on an Olympus BX10 microscope using RATIOTOOL software. Calibration experiments were conducted as in refs. [18,32].

**Patch-clamp recordings in spinal cord slices.** Spinal cord slices (300–400 μm thickness) were prepared from male mice 5–7 weeks of age as in refs. [97,98]. The gramicidin perforated patch method was applied to measure reversal potential for GABA of patched layer-II neurons; 1 mM GABA was applied with a puff-pipette[99,100].

**DARTS assay.** DARTS assay was conducted following[52] using 20 μM KP as "bait" and LC–MS/LC to identify proteins bound to KP. The assay was conducted in both the presence and absence of detergents NP40 and N-dodecyl-b-D-maltoside.

**Molecular dynamics simulation.** Computational binding studies of KP and GSK3ß were conducted using molecular dynamics simulation following[101], using the crystal structure of human GSK3ß, pdb 6V6L.

**Kinome analysis.** Cultured rat primary cortical neurons were treated with either vehicle (DMSO 0.1%) or 1 μM KP for 1 h/24 h. Protein extract was purified, trypsin-digested, and enriched for phosphopeptide via TiO$_2$ resin. The identity and quantity of enriched phosphopeptides were determined using analytical LC–MS/ MS, and then samples were analyzed for differential expression (KP vs. vehicle treatment) for both time points.

**Statistics.** All data are expressed as mean ± SEM. Statistical analysis was conducted using Graphpad Prism9.1 software. Differences between groups were evaluated using two-tailed, unpaired Student's $t$ test (experimental against sham/ vehicle control), or in the case of multiple groups, one-way ANOVA followed by post-hoc Tukey or Dunnett test, following the guidance of the GraphPad program. When applying two-way ANOVA or mixed-model statistics for multiple group comparison, we used post-hoc Bonferroni or Sidak test, following the guidance of the GraphPad program. The criterion for statistical significance was $p < 0.05$.

**Reporting summary.** Further information on research design is available in the Nature Research Reporting Summary linked to this article.

## Data availability
Datasets for all figures and Supplementary figures are contained in a source data file linked to the paper. All source data for screens conducted are contained in Supplementary Data Files S1, S2, S3. Source data are provided with this paper.

## Material availability
All non-commercially available materials will be shared upon request. Source data are provided with this paper.

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

## Acknowledgements

Invaluable comments on the manuscript were provided by Duke University colleagues Drs. James McNamara, Albert LaSpada, Nicole Calakos, Rochelle Schwartz-Bloom and Sidney Simon. We appreciate scientific editing of the manuscript by Dr. David Hauss

(Regeneron). Viral DNA was packaged by the Duke Neurotransgenesis Viral Vector Core, lead-scientist Dr. Boris Kantor. Phospho-proteomics and DARTS analysis were conducted with the expert help of the Duke Proteomics Core Facility, Director Dr. M. Arthur Mosely and Lead-Scientist Dr. Erik Soderblom. KCC2 protein abundance microcapillary assays were carried out by Raybiotech (Norcross, GA). In terms of funding, this work was supported by NIH grants to W.L. (NS066307), Y.C. (DE027454), J.B. (AG056850), Duke Anesthesiology Research Fund to R.-R.J., also by support of the Michael Ross Haffner Foundation (Charlotte, NC) and Duke Neurology internal support to W.L.

## Author contributions

Conducted experiments—M.Y., Y.C., C.Y.J., G.C., K.W., S.C., M.L., P.W., Q.Z., Z.L.W., W.L.; conceptual input—M.Y., Y.C., A.B., J.B., R.-R.J., W.L.; wrote paper—M.Y., Y.C., C.J., G.C., M.L., J.B., R.-R.J., W.L.

## Competing interests

W.L. is a full-time employee of Regeneron Pharmaceuticals (Tarrytown, NY). This affiliation (since April 2021) is not related to the content of this study. Also unrelated to this study, he co-founded TRPblue Inc. (Durham, NC) in December 2017. R.-R.J. is a consultant for Boston Scientific and received research support from this company. These activities are unrelated to this study. Beyond these contacts, the authors declare no competing interests.
