## [Peer Review File · Nature Communications]

Repurposing cancer drugs identifies kenpauellone which ameliorates pathologic pain in preclinical models via normalization of inhibitory neurotransmissionREVIEWER COMMENTS

Reviewer #1 (Remarks to the Author):

In this paper, Yeo et al. reported that kenpaullone (KP), a novel compound that inhibits the GSK3 pathway, has analgesic properties in different preclinical models of chronic pain. They showed that KP-mediated analgesia occurs via upregulation of KCC2 and concurrent increased chloride extrusion from neurons in the dorsal horn of the spinal cord. Although comprehensive and potentially interesting, this study lacks consistency (results are often different across figures) and some methodological details to be accurately replicated by others. Most mechanisms were demonstrated in vitro but not replicated in vivo. In vitro cultures are artificial setups with axotomized neurons and results obtained in vitro may therefore not fully translate what happens in vivo. Supplementary figure 1 is missing and other supplementary figures are mislabeled (there is no Sup Fig 5 for example). Finally, some conclusions and/or statements were not proven or supported by the data presented.

Figure 2

2A: explain why N=10 was needed for the highest dose of KP compared to the lower doses for which N=3-4 only.

2B: need to indicate the real N (not >75)

2C: Compared to Fig 2B, the neuronal chloride concentration for KP is much higher; was the same dose of KP used in 2C compared to 2B? Given the variability in the KP response, this experiment needs to be performed with 3 concurrent groups: veh ctrl, veh+KP and KP+VU.

2D: Ideally the doses used in 2A would have been used in 2D to be consistent. Why use 0.4uM instead of the 0.5uM previously used?

2E: please indicate that these are embryonic cortical neurons in the legend.

2F: again explain the disparity in the number of cells used to analyze Kcc2 vs synaptophysin.

Figure 3: how long after the last dose of systemic KP was behavior measured?

3A: the authors state that this is dose dependent (page 6). Was any statistics done to compare the effect of the 2 doses in CCI? If there is no statistical difference between the 2 doses tested, then the response is not dose dependent.

3B: Same observation as for 3A. The response here is not dose dependent. It's an all or nothing type of response.

3D: results are inconsistent with what is shown in 3C: in 3D the effect of intrathecal KP is reduced at 5h post injection but in 3C there is still an effect at 24h. Is there a rebound of effect between 5h and 24h? Also, in the text the authors say that the effect persisted at 3h and 8h but the histograms show 3h and 24h. Which is it?

Figure 4

4A: How long after CCI were mRNA measured? And how long after the KP injection? What is the rationale for only analyzing KCC2 levels in laminae I-II? KP was injected systemically and according to 4B KCC2 is expressed throughout the spinal cord. Is the effect restricted to the very superficial laminae?

4B: The immunohistochemistry is very poor quality. A panel with a high magnification picture is required to better evaluate what we are looking at. The authors need to show the contralateral side of the PSNL section for one to determine whether there is a decrease after PSNL. The quantification needs to be done comparing ipsi vs contra (not ipsi across different groups). Has this antibody been validated in KCC2 KO? What dilution was it used at.

4C: This diagram is very confusing. It seems that the CCI+KP mice did not develop allodynia at all. Is that so? If so, this is a prevention rather than a reversal. But it's hard to understand what's going on as we don't know how many injections were done ... Was this one injection of KP or 3? Furthermore, these are very low "N" to make any accurate conclusion. A minimum of 5 should be used ...

Figure 5: The figure legend is misleading as what the experiment shows is that both KP and GSK3

inhibitors increase KCC2 expression in vitro. But one may not have anything to do with the other and be completely independent. Similarly, the statement on page 9 that it is the GSK3 inhibitory function of KP that underlies KCC2 upregulation is also not proven. The only way to prove this, is to show that blocking the interaction of KP with GSK3 prevents KCC2 upregulation.

Figure 6:

6C: because both nuclear and cytoplasmic stainings increased after KP incubation, it is difficult to interpret whether the increased staining in the nucleus is a result of translocation rather than an overall increased expression of d-cat. How was the quantification performed? What is the rationale for using a much lower dose of KP (100nM instead of the 0.5uM used in most other experiments)?
- There is mention of Fig 6E and 6F in the text but it should be 6D.

Figure 7: please remove the “non-significant increase” statement from the legend ... if it's not significant then it is not an increase.

7B: Does the k1 deletion (by itself) increase Luc expression compared to WT? and does the k2 deletion by itself decrease it? The authors need to run statistics and compare these groups to determine the contribution of these deletions by themselves.

Figure 8:

8B: The authors must characterize the retrograde transport of AAV9 to the DRG in their own system as Haenraets and colleagues (2017) injected AAV9 intraspinally, not intrathecally. An intrathecal injection is very different from an intraspinal injection.

8D: this result is meaningless as one cannot determine whether KCC2 increased in the neurons that were infected by the AAV-delta-catenin. And the statement that the effect of d-cat(WT) was not significant because it only infected a fraction of neurons makes no sense. Why would the AAV-dcat(WT) infect only a fraction of neurons whereas the AAV-dcat(S276A) would infect a larger number? And if AAV-delta-cat(WT) did not increase KCC2 levels, how do the authors explain its anti-allodynic effect (seen in 8C)?

Supplementary Figures:

S1: this figure is missing so one cannot evaluate the accuracy of what is described in the text.
S7B: why didn't KP enhance luc expression in WT (as shown in Fig 7)?

Text:

Page 6: what does the following mean: “This particular time-course can be interpreted as re-programming of sensitized nociception. Our findings are rather not suggestive of selective analgesic inhibition of a pro-algesic ion channel or receptor.”

Page 9: The statement that inhibition of GSK3 enhances nuclear transfer of delta-cat was not proven. All the authors have reported on is that KP may enhance nuclear translocation of delta-catenin. Whether this occurs through GSK3 inhibition remains to be determined. Similarly, the authors report on the effect of KP on KCC2 expression and the effect of GSK3 inhibitors on KCC2 expression but whether the 2 are linked was not proven. Citing other studies is not sufficient.

Page 10: The statement that beta-catenin plays an ancillary role is not proven ... the authors did not show that beta-catenin needs delta-catenin to enhance expression (or vice versa) ...

Discussion:

-Does delta-cat affect gaba chloride extrusion or does it act via a different mechanism than KP?
-Speculate on what would be the mechanism through which KP reduces cancer pain (if there is no KCC involvement).

Methods:

-Indicate the dilutions used for all primary and secondary antibodies.

Reviewer #3 (Remarks to the Author):

Based on the hypothesis that restoration of *Kcc2* expression levels and improvement of GABA evoked chloride reversal potentials will relieve pathologic pain, Liedtke and colleagues describe a screen for small molecules that increase *Kcc2* transcription using primary mouse cortical neurons harboring a *Kcc2* promoter luciferase reporter. The screen was performed in 24 well plates against 1057 growth regulating compounds from collections curated by NCI, based on the idea that many of them likely impact transcriptional and epigenetic mechanisms. Four hit compounds were identified based on luciferase activity, *Kcc2* transcript levels, and measurements of chloride ion concentration. One of those, kenpaullone (KP), was selected for further analyses.

First the authors show that KP treatment of primary cortical neurons leads to dose responsive increases in *Kcc2* transcripts by RT-PCR, and KCC2 protein by ICC. Moreover, they show that the KCC2 chloride transport inhibitor VU0240551, blocks the effect of KP on $[Cl^-]$, suggesting it is KCC2 dependent. Some of these results are confirmed in primary human neurons where they show dose dependent *Kcc2* transcript increases, as well as KCC2 protein increase by ICC that correlates with increased synaptophysin expression. They then go on to show that in vivo, using two pain models in mice- nerve constriction injury, and bone cancer- that KP acts as an analgesic for mechanical withdrawal and allodynia, again in a KCC2 dependent manner based on co-administration of VU0240551. These in vivo studies are followed up with an ex-vivo examination of dissected spinal cord dorsal horn (SCDH), where increased *Kcc2* transcription by RT-PCR, and KCC2 protein by ICC is demonstrated. In addition, perforated patch recordings were performed on slices to demonstrate normalization of EGABA in KP treated animals. Finally, the authors perform a series of experiments, including *Kcc2* promoter crunching, aimed at understanding the mechanism of action of KP. These experiments lead to the proposal that KP inhibits GSK3b (not CDKs), resulting in decreased S276 phosphorylation, stabilization and nuclear translocation of delta catenin, and upregulation of *Kcc2* via the delta catenin-Kaiso pathway. In a final in vivo experiment, the authors show that overexpression of S276 delta catenin in SCDH via AAV9, phenocopies KP treatment.

This manuscript describes extensive testing of the hypothesis that regulation of *Kcc2* transcript levels by a GSK3b dependent pathway represents a potential target for pain management, an important, unmet medical need. The manuscript is of general interest, especially to those in the field of pain management. However, in its current form the manuscript is not recommended for publication in Nature Communications. If the authors could address the concerns below, it is recommended that the manuscript be reconsidered.

Major comments:

- 1) The authors are proposing a direct relationship between KP treatment and KCC2 protein levels. However, the manuscript is entirely dependent on ICC to establish that KP increases KCC2 protein levels. Western blots need to be included at least for primary mouse cortical neurons, to allow quantification of KCC2 protein levels over time in response to KP.
- 2) In addition, protein levels assessed by ICC in Fig. 2A should be quantified, and if possible, the spinal cord sections shown in Fig. 4B should include a control marker not affected by KP that can be used to normalize KCC2 expression between animals.
- 3) More information needs to be provided regarding quality control for the screen that will allow the reader to assess assay robustness. What was the coefficient of variation across wells? A scatter plot of the results including vehicle controls should be presented, ideally as Z-scores, or percent change from vehicle controls, so inter-plate results could be compared. What was the relative increase in KCC2 expression compared with vehicle controls? The authors note a Z'-factor of 0.94. How many wells were used to calculate the Z-factor? Was the Z-factor calculated for each plate?
- 4) Full KP dose response experiments traversing multiple logs should be performed for luciferase, and ideally mRNA levels, at least in primary mouse cortical neurons, to establish an EC50. In addition, inclusion of similar data for the other 3 hits identified would strengthen the manuscript.

5) The KCC2 and synaptophysin co-stainings (Fig. 2E-F) are overinterpreted. Why is synaptophysin so weak? Is the culture very immature? Are the authors proposing that KP induces maturation, and that its effect on Kcc2 transcription is indirect? An analysis of the effects of KP on primary human neurons that are cultured until they are mature enough to express synaptophysin, should be performed, as it would better address the question of whether KP is increasing KCC2 in human neurons in a more relevant context. In addition, other markers of synaptic maturation need to be included to support the claim on p.6 that “in human primary cortical neurons we found enhanced synaptic maturation and increased KCC2 expression, most likely causally linked”.

6) The increased nuclear cytoplasmic localization shown in Fig. 6C and D is not clear from the ICC images. How was the quantification performed? Is it simply the average intensity of nuclear expression? A ratio of nuclear to cytoplasmic expression levels should be included. This would address whether the ratio has increased or whether overall levels of delta catenin have increased, and also allow for normalization between cultures.

7) There is a glancing reference to a similar screen that was performed in the context of Rett syndrome (Tang, X. et al., 2019). One of the confirmed KCC2 inducing hits that was investigated in this study was the GSK3b inhibitor BIO, in addition to follow-up with the GSK3b inhibitor TWS-119. This reference deserves more discussion as it considerably strengthens the authors' identification of a GSK3b inhibitor, KP, that increases Kcc2 expression.

Minor comments:

1) The authors have included a large amount of experimental data, so much so that there are many points throughout the manuscript where the reader needs to search for basic information (e.g. days of treatment), making the manuscript difficult to read. A careful editing of the results section to ensure inclusion of information that the reader requires to assess the results should be performed. If space constraint is a problem, the description of the screen could easily be moved to the supplementary data section.

2) Validation of KCC2 antibodies is unclear on p. 42 “Anti-KCC2 primary antibodies were validated with developing rat primary cortical neurons; we observed an increase in staining pattern that tightly matched increase of Kcc2 mRNA expression.” Based on what assay? ICC? Western?

3) The manuscript would be strengthened by including a discussion of whether there are GSK3b inhibitors in clinical development or trials.

4) All supplementary figure references need to be edited. For example, there is no supp Fig. S1 or S5, and there is inconsistent naming (Suppl Fig. vs Fig. S), etc., too many instances to list here.

Please see point-by-point responses to reviewers for relevant detail on the following pages:

REVIEWER COMMENTS

Reviewer #1 (Remarks to the Author):

In this paper, Yeo et al. reported that kenpaullone (KP), a novel compound that inhibits the GSK3 pathway, has analgesic properties in different preclinical models of chronic pain. They showed that KP-mediated analgesia occurs via upregulation of KCC2 and concurrent increased chloride extrusion from neurons in the dorsal horn of the spinal cord. Although comprehensive and potentially interesting, this study lacks consistency (results are often different across figures) and some methodological details to be accurately replicated by others.

We thank the reviewer for his/her in-depth review and the encouraging evaluation.

We have thoroughly worked on the issues pertaining to consistency and methodological detail.

Most mechanisms were demonstrated in vitro but not replicated in vivo. In vitro cultures are artificial setups with axotomized neurons and results obtained in vitro may therefore not fully translate what happens in vivo.

We fully agree with the reviewer that in vitro cultures have limitations re translation to whole animal experiments. However, our in vitro system is very useful for initial drug screening and mechanistic studies. Importantly, we have validated our key findings in vivo, including 1) regulation of gene expression of *Kcc2* in the spinal cord (Fig. 4a-b), 2) in-lockstep regulation of KCC2 chloride extrusion function in spinal cord dorsal horn neurons, by measuring their E_{GABA} (Fig. 4d), 3) pain behaviors in animal models of neuropathic pain and cancer pain following KP treatment (Figs. 3, 4b), and 4) gene therapy in vivo, directed to the spinal cord, to enhance the pathway down-stream of GSK3 β that we have verified in primary neurons, δ -catenin with resulting enhanced expression of *Kcc2* (Fig. 8). We have discussed the limitations of primary neurons in culture.

Supplementary figure 1 is missing and other supplementary figures are mislabeled (there is no Sup Fig 5 for example). Finally, some conclusions and/or statements were not proven or supported by the data presented.

Original Suppl Fig. 1 was Suppl Dataset 1. We apologize for the confusion in naming Supplementary Files. This issue has now been updated throughout the revised ms.

Figure 2

2A: explain why N=10 was needed for the highest dose of KP compared to the lower doses for which N=3-4 only.

Iterations of the high-dose experiment were conducted on multiple occasions, lower dose only n=3-4.

2B: need to indicate the real N (not >75)

Addressed as suggested by the reviewer, n=75 and n=76, now shown in the Figure Legend of Fig. 2d.

2C: Compared to Fig 2B, the neuronal chloride concentration for KP is much higher; was the same dose of KP used in 2C compared to 2B? Given the variability in the KP response, this experiment needs to be performed with 3 concurrent groups: veh ctrl, veh+KP and KP+VU.

We profoundly apologize for mislabeling former subpanel 2c.

The left-hand black bar and black dots stand for vehicle control + VU, the red/purple bar and dots stand for KP + VU.

The color-coding is in keeping with the entire Fig. 2, the written legend remains appropriate.

This is an important control experiment for previous panel 2B, now 2d. We have also clarified that the same doses of KP were used for 2d and 2e.

In other words, if KCC2 transport function is inhibited, this leads to a high $[Cl^-]_i \geq 120mM$, irrespective of whether *Kcc2* gene expression and subsequently KCC2 transporter function were enhanced by KP or not. Of note, this experiment also illustrates the impact of endogenous KCC2 function in primary cortical neurons, with approximately doubling of $[Cl^-]_i$ in response to VU.

- These findings in turn lead to the conclusion that KP's chloride lowering effect relies on KCC2 chloride extruding transport function.

2D: Ideally the doses used in 2A would have been used in 2D to be consistent. Why use 0.4uM instead of the 0.5uM previously used?

The reviewer is correct, we are in agreement. – The discrepancy is due to a communication glitch between two experimental sites (WL's lab at Duke; JB's lab at UCI).

2E: please indicate that these are embryonic cortical neurons in the legend.

We are now referring to the human cortical neurons as "primary human fetal cortical neurons".

2F: again explain the disparity in the number of cells used to analyze Kcc2 vs synaptophysin.

The legends have now been edited as suggested. The numbers shown are not the number of cells but the numbers of the images analyzed. The apparent disparity in the number of images is due to a technical issue - some images were double-labeled for KCC2 and synaptophysin, and the others were labeled for KCC2 and NeuN (not shown, mentioned in "Detailed Methods"). Thus a higher number of images were included in the statistical analysis of KCC2 labeling intensity compared to synaptophysin labeling intensity. We have edited the figure legend to clarify this issue (see revised legend Fig. 2h).

Figure 3: how long after the last dose of systemic KP was behavior measured?

Behavior was measured 16h after the last dose of systemic KP, now indicated in the revised ms.

3A: the authors state that this is dose dependent (page 6). Was any statistics done to compare the effect of the 2 doses in CCI? If there is no statistical difference between the 2 doses tested, then the response is not dose dependent.

We examined our statistical analyses again. In the revised figure legend, we are now making reference to our result that there is dose-dependency for the 0 - 10mg - 30mg KP doses according to linear regression measurements. We recorded p-values of $p=0.021$ (d4), $p=0.0004$ (d7), $p<0.0001$ (d11 and d14). In Suppl Fig. 3a, we are now illustrating our result of significant dose-dependent analgesia when applying a repeated measures mixed effects model where all time-points were simultaneously included in the analysis.

3B: Same observation as for 3A. The response here is not dose dependent. It's an all or nothing type of response.

We respectfully differ with the reviewer. The statistically significant difference between 10mg/kg and 30mg/kg KP groups at time-points 10d and 14d suggests that the analgesic effect is associated with dose and therefore can be described as "dose-dependent".

3D: results are inconsistent with what is shown in 3C: in 3D the effect of intrathecal KP is reduced at 5h post injection but in 3C there is still an effect at 24h. Is there a rebound of effect between 5h and 24h? Also, in the text the authors say that the effect persisted at 3h and 8h but the histograms show 3h and 24h. Which is it?

We thank the reviewer for the insightful discussion. There are some differences in baselines after nerve injury in different cohorts of animals.

When comparing 3d with 3c, one has to realize that 3d shows a one-shot experiment, whereas 3c shows the results of intrathecal injections on 3 consecutive days. Importantly, the second and third days of injection differ from the first in that the injection of KP proves to be more potent and longer lasting. Upon first intrathecal injection on d6, there is a tendency toward increased effectiveness, but differences between KP and vehicle are not significant. On the other hand, 3d shows first intrathecal injection on d7, with mechanical threshold becoming less sensitive from 0.28 to 0.68 in response to intrathecal KP, 3h post i.t-injection, and from 0.28 to 0.6 at 4h. These differences are statistically highly significant, now also indicated in revised subpanel 3d.

Back to the reviewer's question: 3c and 3d are not directly comparable because repeated intrathecal injection of KP is associated with increased effectiveness and duration of action. The one-time injection on d7 as shown in 3d was not done in the experiment shown in 3c.

We did not observe any rebound of effect. Upon repeat injection, there is a significant effect at 24h, but this is less effective than at 3h post-i.t injection.

The effect is present at 3h and persists until 24h. We have now corrected the text accordingly, apologize for the incorrect reference to 8h.

Figure 4

Upfront, we like to reiterate that the main objective of Fig. 4 is to demonstrate that attenuated *Kcc2* gene expression in the spinal cord dorsal horn superficial layers (= pain-relaying layers), which was a result of peripheral nerve injury that evokes pain, can be repaired by treatment with KP. KP treatment renormalizes *Kcc2* expression in superficial layers I-II of the spinal cord dorsal horn, known to be key for pain relay. Here, in pain relaying neurons, less negative GABA reversal potential, which was caused by peripheral nerve injury, is renormalized to physiologically more negative values by KP treatment.

4A: How long after CCI were mRNA measured?
miRNA was measured at 7d after nerve injury

And how long after the KP injection?
KP was injected daily after PSNL nerve injury. SCDH was microdissected 8h after the last injection of KP at a dose of 10mg/kg on d7.

What is the rationale for only analyzing KCC2 levels in laminae I-II?
Because laminae I-II are critical for nociceptive transmission and do show robust KCC2 expression (Fig. 4b). Please also see previous studies in refs. 3, 6 and 45 of our revised ms.

KP was injected systemically and according to 4B KCC2 is expressed throughout the spinal cord. Is the effect restricted to the very superficial laminae?
Our observation of KCC2 expression throughout the entire spinal cord gray matter is consistent with the literature (Neuroscience. 2008, 152(2):502-10; J Physiol. 2008, 586(23):5701-15; Nat Commun. 2020, 11(1):393). The effect in response to KP treatment is not restricted to the superficial layers of the dorsal horn, however the superficial layers are the critical layers for relay of afferent nociceptive signals and show dramatic changes after nerve injury and KP treatment (Fig. 4a, b, d).

4B: The immunohistochemistry is very poor quality. A panel with a high magnification picture is required to better evaluate what we are looking at.
We have replaced the images; please see revised Fig. 4b.

The authors need to show the contralateral side of the PSNL section for one to determine whether there is a decrease after PSNL. The quantification needs to be done comparing ipsi vs contra (not ipsi across different groups).

We appreciate the reviewer's thoughtful comment.

The main thrust of Fig. 4 is demonstration of the e-phys correlate in pain-transmitting neurons in the spinal cord dorsal horn. We intended to demonstrate consequences of peripheral nerve injury by measuring a more positive, hyperexcitable E-GABA in these neurons, and then rescue of this neurophysiologic pain correlate by treatment with KP, supported by observations of renormalized gene expression of *Kcc2* in the superficial layers of the spinal cord dorsal horn, and by rescue of behavioral sensitization. The method of choice to conduct this experiment is comparison between groups, sham vs. injury vs injury plus KP, and measuring E-GABA from superficial dorsal spinal pain relaying neurons on the side of injury to permit the group-by-group comparison. Gene-expression studies and nocifensive behavior metrics are ancillary to e-phys, and we have therefore rigorously adhered to ipsilateral measurements throughout this experiment.

Ipsi- vs contralateral measurements are nevertheless interesting, but we refer here to future investigations looking into this worthy approach. Such studies can shed light onto mechanisms how ipsilateral lesions affect contralateral physiology, gene expression and behavior, as suggested by Sotgiu & Biella, doi: 10.1016/s0304-3940(98)00589-8, Won & Lee, doi: 10.1155/2015/438319. Epub 2015 Sep 28, and as well-known from clinical practice.

Has this antibody been validated in KCC2 KO?

We were not able to gain access to *Kcc2*^{-/-} cultured neurons.

We used the Millipore antibody 07-432, raised against the C-terminus of the KCC2 protein, also used and validated by other investigators, which we have now referenced.

We validated this antibody by comparing DRG neurons (negative control) vs matured cortical neurons

(KCC2-expressing), and HEK293t cell line (KCC2 negative when control transfected, KCC2 robust expression upon transfection with KCC2 cDNA).

This new information is now contained in Supplementary Fig. 2 of the revised manuscript.

What dilution was it used at.

Addressed as suggested by the reviewer, it was used at 1:2000 (See Supplementary Table; 1:400 ICC)

4C: This diagram is very confusing.

We improved the clarity of the diagram by displaying it as bar diagram with individual data points as dots, for the three groups, the two time-points and brackets to display significance levels.

It seems that the CCI+KP mice did not develop allodynia at all. Is that so?

Yes this is an important point.

If so, this is a prevention rather than a reversal.

Yes, this is prevention or "early intervention".

But it's hard to understand what's going on as we don't know how many injections were done.

KP was injected once before injury, and thereafter daily injections post-injury for a total of 2 post-injury injections. This has been clarified in the revised ms.

Was this one injection of KP or 3?

KP was injected 3 times, once before PSNL, two times post-injury.

Furthermore, these are very low "N" to make any accurate conclusion. A minimum of 5 should be used.

As mentioned above, we now have increased n=5 animals per group, with a sample size of 9-12 neurons per group. In each animal, we recorded from 1-3 neurons.

Figure 5: The figure legend is misleading as what the experiment shows is that both KP and GSK3 inhibitors increase KCC2 expression in vitro.

Addressed as suggested by the reviewer, we changed the title to "*Kenpaullone and GSK3-inhibitors, not CDK-inhibitors, increase Kcc2 expression in cultured central neurons*".

But one may not have anything to do with the other and be completely independent.

We acknowledge this possibility.

KP was shown to inhibit GSK3 β with an IC₅₀ of reportedly as low as 23nM (*Curr Top Med Chem* 11, 1320-1332, 2011). Indeed, of all known targets of KP, its GSK3 β -inhibitory potency is the highest, as measured by reported IC₅₀ values. The next potentially inhibited targets of KP, CDKs, are inhibited at several hundred nanomolar potency (*Cancer Research* 59, 2566-2569, 1999). It is possible that KP may increase KCC2 expression via GSK3-independent mechanisms. However, we have demonstrated that GSK3 β downstream signaling through δ -cat is sufficient to regulate KCC2 expression in the spinal cord and reverse neuropathic pain (Fig. 8).

Similarly, the statement on page 9 that it is the GSK3 inhibitory function of KP that underlies KCC2 upregulation is also not proven.

We are accommodating the reviewer's argument by changing how we express our conclusion.

We have attempted to make more clear a central message of our experiments, which rests on the following 6 arguments.

1) KP is a pleiotropic kinase inhibitor. Of its identified and verified kinase targets, GSK3 β is inhibited at the lowest IC₅₀, at 23nM, up to 75nM in different assays. Other inhibited kinases such as CDKs have IC₅₀ values at several hundred nanomolar and higher (*Curr Top Med Chem* 11, 1320-1332,2011; *J Med Chem* 42, 2909-2919, 1999).

2) We demonstrate direct binding of KP to GSK3 β in primary neurons, a novel finding. Whereas GSK3 β was readily identified, other known kinase targets of KP were not retrieved in our unbiased binding assay (Fig. 6A, Suppl File S2).

3) Our experimental evidence suggests that δ -cat functions down-stream of GSK3 β as its kinase target in primary neurons. Our evidence is in agreement with the established record of δ -cat as a direct kinase target of GSK3. Experimentally, we identify the relevant phosphorylation site of δ -cat, a novel finding in neurons.

4) δ -cat phosphorylation at S259 (rat; its equivalent S276 in human) favors cytoplasmic degradation, and δ -cat that cannot be phosphorylated at this site is more likely to traffic to the nucleus.

5) In the nucleus of the neuron, δ -cat enhances expression of *Kcc2* by binding to two Kaiso sites that bracket the transcriptional start site of the *Kcc2* gene (Fig 7).

6) From the primary neuron, we take this mechanism back into the live animal and demonstrate that spinal transgenesis with δ -cat functions in analgesic manner, with an S276A mutant more potent than WT (Fig 8).

These 6 key points in ensemble constitute suggestive evidence that KP enhances *Kcc2* gene expression by its GSK3 β inhibitory effect.

The only way to prove this, is to show that blocking the interaction of KP with GSK3 prevents KCC2 upregulation.

We were not able to prove this in *GSK3 β ^{-/-}* cells because *GSK3 β ^{-/-}* mice are not viable, and deriving cell lines from very early embryos has not been successful because of failure of such cells to divide and thrive in culture.

We resorted instead to two strategies, as shown in Suppl Fig. 5.

1) We co-applied KP to primary cortical neurons together with the CHIR99201 compound, a mono-selective GSK3 inhibitor and a widely used reagent (*PLoS ONE*, 8, e60148, 2013). We did not observe that co-application led to additive effects on *Kcc2* expression, whereby both approaches by themselves increased *Kcc2* expression as we saw before.

This was also true in N2a neuronal cells, which express moderate levels of *Kcc2*, share gene regulatory mechanisms of *Kcc2* with primary neurons as we have shown in our initial paper on *Kcc2* gene regulation.

2) We also used azakenpauillone in primary cortical neurons and observed significantly increased *Kcc2* gene expression. Of note, azakenpauillone is more potently targeting GSK3 β (EC50 18nM) than kenpauillone and has 100-fold reduced potency vs other kinases (*Bioorg Med Chem Lett* 14, 413-416, 2004).

Figure 6:

6C: because both nuclear and cytoplasmic stainings increased after KP incubation, it is difficult to interpret whether the increased staining in the nucleus is a result of translocation rather than an overall increased expression of δ -cat. How was the quantification performed?

This is detailed in our **Methods/Detailed Methods** sections.

Quantification was performed on confocal micrographs so that immunoreactivity in the nucleus was recorded strictly separate from cytoplasmic immunoreactivity. Nuclear immunoreactivity was measured as relative value, normalized for cytoplasmic immunoreactivity. We measured cytoplasmic immunoreactivity and it remained unchanged between groups.

This method enabled us to draw conclusions about abundance of δ -cat in the nucleus in response to KP treatment.

What is the rationale for using a much lower dose of KP (100nM instead of the 0.5 μ M used in most other experiments)?

100nM was used and yielded a significant level of nuclear transfer. This suggests considerable potency of KP in some assays, in keeping with upregulation of *Kcc2* expression which we observed also at this concentration. Furthermore, KP was also used at 100nM for experiments shown in Fig. 7. However, in an attempt to conduct the experiment for Fig. 6 at 500nM concentration of KP for the revision, we discovered that the supply chain of the primary anti- δ -cat antibody was disrupted during the pandemic.

- There is mention of Fig 6E and 6F in the text but it should be 6D.

The reviewer is correct, we apologize. We regret this lack of oversight. We have now rectified the error. Figure 7: please remove the “non-significant increase” statement from the legend ... if it's not significant then it is not an increase.

We have rephrased it, as suggested by the reviewer.

7B: Does the k1 deletion (by itself) increase Luc expression compared to WT?

Yes, it does, as shown in Fig. 7b (lower bar diagram, left-hand side, white and gray bars).

and does the k2 deletion by itself decrease it?

Yes, it does, same explanation.

The authors need to run statistics and compare these groups to determine the contribution of these deletions by themselves.

We did; the increases and decreases of activity of the *Kcc2* promoter were statistically significant. This is indicated in Fig. 7b (left-hand bar diagram), please note significance levels indicated with brackets. Since δ -cat binding to the *Kcc2* promoter is a central finding of our mechanistic studies, we have included it in the title of the Figure legend for Fig. 7 "*Delta-catenin binds to the proximal Kcc2 promoter at two Kaiso sites, regulated by Kenpaullone*". The contribution of each Kaiso site and their combined impact is then elucidated in the deletion ("promoter bashing") studies in Figure 7b.

Figure 8:

8B: The authors must characterize the retrograde transport of AAV9 to the DRG in their own system as Haenraets and colleagues (2017) injected AAV9 intraspinally, not intrathecally. An intrathecal injection is very different from an intraspinal injection.

We thank the reviewer for pointing out this important difference, we fully agree.

We have conducted a new experiment by i.t injection. Our results show that retrograde transport of AAV9-tdTomato to the DRG is virtually non-detectable on d4 and with very low expression on d13 post-injection. In contrast, many spinal cord cells are labeled at both time points (Fig. 8b). We conclude that early transduction of the spinal cord and SCDH with the fluorescent marker indicates early and appreciable promoter activity in central neurons vs low activity in peripheral neurons. This in turn suggests a predominantly central action of the δ -cat transgenes. It cannot be excluded that at the later time points, d(15-18), there can be some low-level expression of the transgene in the DRG.

8D: this result is meaningless as one cannot determine whether KCC2 increased in the neurons that were infected by the AAV-delta-catenin. And the statement that the effect of δ -cat(WT) was not significant because it only infected a fraction of neurons makes no sense. Why would the AAV-dcat(WT) infect only a fraction of neurons whereas the AAV-dcat(S276A) would infect a larger number? And if AAV-delta-cat(WT) did not increase KCC2 levels, how do the authors explain its anti-allodynic effect (seen in 8C)?

We have rephrased our conclusions that relate to this figure.

Our results with the tdTomato reporter gene suggest that the key pain relay neurons in the SCDH do not get virally transduced to (close to) 100% effectiveness for appreciable expression. The viral transduction efficiency is lower, as shown in Figure 8B. The observed level of transduction efficiency does not differ between WT and S276A transgene. The S276A δ -cat transgene leads to significant analgesia on days 7, 10, 14 and 18, whereas this is true for the WT δ -cat transgene only on days 10 and 18. Overall this means that the S276A transgene functions consistently analgesic, whereas the WT transgene is less consistently analgesic.

Our interpretation is that this difference can be explained by the S276A transgene trafficking to the neurons' nucleus more likely than its WT counterpart, in turn up-regulating *Kcc2* gene expression more effectively. This is based on findings in central neurons as shown in Figs. 6-7. Importantly, mRNA quantification from the spinal cord dorsal horn is based on homogenized tissue, thus diluting transduced neurons with non-transduced, so that in turn we were measuring a lower effect than that accomplished in 100% virally transduced neurons.

The up-regulation of *Kcc2* gene expression is significant for the phosphorylation-incompetent S276A mutant which is more potent at nuclear trafficking, even though non-transduced neurons diluted this

effect. With similar % of virally transduced neurons for S276A and WT, the WT transgene's effect with less nuclear translocation gets diluted so much so that there is no significant increase. We think these results are meaningful.

We believe our findings are suggestive in the context of what we have learned about δ -cat's enhancing impact on *Kcc2* gene expression in case of δ -cat translocation to the nucleus. These insights have been derived from cellular models of primary central neurons, as shown in Figs. 5-7 and their respective Supplementary Figures 5-7 and Supplementary Datasets S2-3.

Supplementary Figures:

S1: this figure is missing so one cannot evaluate the accuracy of what is described in the text.

Again, we apologize for the confusion.

“S1” is represented by “Supplementary File S1, Excel sheet #1: Compound screening results”.

S7B: why didn't KP enhance luc expression in WT (as shown in Fig 7)?

In Figure 7B and in Supplementary Figure 7B, in WT, KP does not increase LUC activity to significant degree, as shown in both figures.

Text:

Page 6: what does the following mean: “This particular time-course can be interpreted as re-programming of sensitized nociception. Our findings are rather not suggestive of selective analgesic inhibition of a pro-algesic ion channel or receptor.”

We have now clarified this statement on page-break 6/7.

Page 9: The statement that inhibition of GSK3 enhances nuclear transfer of delta-cat was not proven. All the authors have reported on is that KP may enhance nuclear translocation of delta-catenin.

We have accommodated the reviewer's concern in our revised ms. throughout

See also our more detailed statement in response to the reviewer's comments on Figure 5, our 6 central arguments.

Whether this occurs through GSK3 inhibition remains to be determined. Similarly, the authors report on the effect of KP on KCC2 expression and the effect of GSK3 inhibitors on KCC2 expression but whether the 2 are linked was not proven. Citing other studies is not sufficient.

We did demonstrate direct binding of KP to GSK3 β in CNS neurons, not to any CDK that has ever been found a KP target, by using an unbiased screening method (Fig. 6a; Suppl Dataset 2).

GSK3 β is the kinase targeted for KP with the lowest IC₅₀. δ -cat is an established GSK3 β kinase target.

We then demonstrate that a specific δ -cat phosphorylation site that we identify - for the first time in neurons - is significantly less phosphorylated at the 48h time-point when treating with KP.

This finding is based on data shown in the Supplementary Files S2 and S3. Data in each file are rather voluminous, but our step-by-step analysis has the following logic

1) KP binds to GSK3 β , not any other known KP pharmacologic targets.

2) Our DARTS assay data allow conclusion as in 1) in primary neurons. These findings are extended by our new result that KP may bind to GSK3 β in its ATP binding pocket as other GSK3 β inhibitors (Fig. 6a. Suppl Fig. 6a)

3) δ -cat, known bona-fide phosphorylation target of GSK3 β , is the highest-priority phosphorylation target when treating with KP. We identify the specific phosphorylation site in δ -cat.

These findings can only be reconciled with KP inhibiting GSK3 β as underlying mechanism.

4) Multiple known selective and specific inhibitors of GSK3 mimic the effect of KP on upregulation of *Kcc2* gene expression. Through experimentation that we have conducted for this revision, we learn that. (i) The selective GSK3 inhibitor CHIR99201 enhances *Kcc2* gene expression to similar degree as KP. When adding KP to CHIR99201 there is no additional upregulation of *Kcc2* gene expression. (ii) structurally highly similar to KP, aza-KP, which has an IC₅₀ vs GSK3 β of 18nM and has 100-fold less

activity against CDK1, significantly upregulates gene expression of *Kcc2* at 20nM.

5) A very recent reference supports this concept, as highlighted by Reviewer #2 (*One of the confirmed KCC2 inducing hits that was investigated in this study was the GSK3b inhibitor BIO, in addition to follow-up with the GSK3b inhibitor TWS-119. This reference deserves more discussion as it considerably strengthens the authors' identification of a GSK3b inhibitor, KP, that increases Kcc2 expression.*). We have accommodated R2's suggestion in this respect, and agree with R2 that this makes our conclusions stronger.

Page 10: The statement that beta-catenin plays an ancillary role is not proven ... the authors did not show that beta-catenin needs delta-catenin to enhance expression (or vice versa) ...

We have now added new data showing that when using a β -cat-inhibitory compound, ICG-001, this decreases *Kcc2* gene expression significantly, shown in Suppl Fig. 7b.

We have accommodated the reviewer's critique in the revised ms. when referring to β -catenin.

We believe that our Supplementary data support an ancillary role of β -cat in enhancing *Kcc2* gene expression.

Overall, referring to β -cat's ancillary role does not mean that this role must be critical, and we have rephrased our comments.

Discussion:

-Does delta-cat affect gaba chloride extrusion or does it act via a different mechanism than KP?

Analgesia in response to i.t transgenesis with δ -cat transgenes is protracted over days to weeks. This protracted time course argues in favor of a slow-and-steady gene-regulatory effect, which we demonstrate in the spinal cord dorsal horn for *Kcc2* mRNA expression in response to the δ -cat(S276A) transgene (Fig. 8c), also in a neuronal cell line for the δ -cat(S276A) and δ -cat(WT) transgene (Fig. 8a). Our experimental evidence suggests both KP and δ -cat to enhance gene expression of *Kcc2*. For KP, our evidence supports this to be caused by inhibition of GSK3 β . δ -cat is the direct kinase target of GSK3 β . The rationale of our approach is to test the next signaling step after GSK3 β in live animals, and address the question whether we see similar effects with respect to behavioral metrics of pain. For KP, we present evidence that it is not a KCC2-mediated chloride extrusion enhancer in early time points (10 and 20 min, Suppl Fig. 2d-f). We cannot exclude that δ -cat, in addition to its *Kcc2* gene-expression-enhancing effect, functions as a chloride extrusion enhancer in the late time-points (3h and 8h). However, the time course of the analgesic effects of δ -cat transgenesis ($\geq 7d$) argues against this. It is our opinion that decisively addressing this interesting question by experimentation goes beyond the scope of this revision.

-Speculate on what would be the mechanism through which KP reduces cancer pain (if there is no KCC involvement).

We have added to our Discussion, as encouraged by the reviewer.

Our data support an analgesic effect of KP on cancer pain via its *Kcc2* gene expression enhancing effect in the central nervous system, not via a growth-regulatory effect on malignantly growing cells, as our bone destruction score was not altered by KP at the same dose (Suppl Fig. 3b). However, higher doses of KP (which are safe in rodents) might do both, and this makes the compound more attractive. We did mention that at the translational level, combining *Kcc2* gene-expression enhancing approaches with other analgesic strategies could be particularly beneficial, as also echoed in our reference #80 by Lorenzo-et-al. The reviewer's suggestion makes for the excellent argument that for cancer pain, combining KP with other growth-regulatory compounds that can attenuate pain, e.g valproic acid as an HDAC-inhibitor, might enhance KP's cancer pain analgesic effects.

Methods:

Indicate the dilutions used for all primary and secondary antibodies. **Addressed as suggested (Suppl Table).**

Reviewer #3 (Remarks to the Author):

Based on the hypothesis that restoration of Kcc2 expression levels and improvement of GABA evoked chloride reversal potentials will relieve pathologic pain, Liedtke and colleagues describe a screen for small molecules that increase Kcc2 transcription using primary mouse cortical neurons harboring a Kcc2 promoter luciferase reporter. The screen was performed in 24 well plates against 1057 growth regulating compounds from collections curated by NCI, based on the idea that many of them likely impact transcriptional and epigenetic mechanisms. Four hit compounds were identified based on luciferase activity, Kcc2 transcript levels, and measurements of chloride ion concentration. One of those, kenpaullone (KP), was selected for further analyses.

First the authors show that KP treatment of primary cortical neurons leads to dose responsive increases in Kcc2 transcripts by RT-PCR, and KCC2 protein by ICC. Moreover, they show that the KCC2 chloride transport inhibitor VU0240551, blocks the effect of KP on $[Cl^-]$, suggesting it is KCC2 dependent. Some of these results are confirmed in primary human neurons where they show dose dependent Kcc2 transcript increases, as well as KCC2 protein increase by ICC that correlates with increased synaptophysin expression. They then go on to show that in vivo, using two pain models in mice- nerve constriction injury, and bone cancer- that KP acts as an analgesic for mechanical withdrawal and allodynia, again in a KCC2 dependent manner based on co-administration of VU0240551. These in vivo studies are followed up with an ex-vivo examination of dissected spinal cord dorsal horn (SCDH), where increased Kcc2 transcription by RT-PCR, and KCC2 protein by ICC is demonstrated. In addition, perforated patch recordings were performed on slices to demonstrate normalization of EGABA in KP treated animals. Finally, the authors perform a series of experiments, including Kcc2 promoter crunching, aimed at understanding the mechanism of action of KP. These experiments lead to the proposal that KP inhibits GSK3b (not CDKs), resulting in decreased S276 phosphorylation, stabilization and nuclear translocation of delta catenin, and upregulation of Kcc2 via the delta catenin-Kaiso pathway. In a final in vivo experiment, the authors show that overexpression of S276 delta catenin in SCDH via AAV9, phenocopies KP treatment.

This manuscript describes extensive testing of the hypothesis that regulation of Kcc2 transcript levels by a GSK3b dependent pathway represents a potential target for pain management, an important, unmet medical need. The manuscript is of general interest, especially to those in the field of pain management.

We appreciate the reviewer's positive evaluation.

However, in its current form the manuscript is not recommended for publication in Nature Communications. If the authors could address the concerns below, it is recommended that the manuscript be reconsidered.

We thank the reviewer for the constructive comments which we believe have greatly improved this study. We hope that our revised ms can now be reconsidered.

Major comments:

1) The authors are proposing a direct relationship between KP treatment and KCC2 protein levels. However, the manuscript is entirely dependent on ICC to establish that KP increases KCC2 protein levels. Western blots need to be included at least for primary mouse cortical neurons, to allow quantification of KCC2 protein levels over time in response to KP.

We have now determined abundance of KCC2 protein in neuronal cultures by conducting microfluidic electrophoretic separation of protein extract from cultured rat primary neurons with subsequent immunodetection. This leads to a result very similar to Western blotting. The result shows a specific band of KCC2 at a molecular weight of approximately 150kDa, appearing there also in mouse brain protein extract, using an anti-KCC2 antibody against the N-terminus which has previously been validated. We demonstrate significant upregulation of KCC2 in response to KP treatment (0.5 μ M). These results are now included into revised Fig. 2 and Suppl Fig. 2.

2) In addition, protein levels assessed by ICC in Fig. 2A should be quantified, and if possible, the spinal cord sections shown in Fig. 4B should include a control marker not affected by KP that can be used to normalize KCC2 expression between animals.

We have quantified ICC as shown now in Fig. 2b.

Re Fig. 4b we did not find suitable control markers. We replaced the images of Fig. 4b with higher

resolution. We also validated the specificity of KCC2 antibody (Suppl Fig. 2a), also see response to R1.

3) More information needs to be provided regarding quality control for the screen that will allow the reader to assess assay robustness.

What was the coefficient of variation across wells?

We calculated the coefficient of variation as 6% for vehicle control and 5.3% for positive control.

A scatter plot of the results including vehicle controls should be presented, ideally as Z-scores, or percent change from vehicle controls, so inter-plate results could be compared.

We now show a scatter plot of 30 experiments in Suppl Fig. 1a-b. These 30 experiments were the primary screen. We show the reporter gene readout of the vehicle control vs the positive control, and are also showing it as -fold increase of positive control vs vehicle control in a dot-plot.

Importantly, our screen was conducted using primary cortical neurons derived from late rat embryos. With this type of cells for a screen, their source and resulting batch-to-batch (that is rat pregnancy to rat pregnancy) variation it is worth remembering. Per screening experiment we screened between 1-3 plates, depending on cell yield/availability. We know that maturation of neurons as they are taken into culture from individual pregnancies can be variable, therefore we believe that our efforts at accomplishing consistent plate-to-plate drive on the *Kcc2* promoter have yielded quite satisfactory outcome, when looking at the raw RLU output. Number of wells per control condition were between 2 and 4 per plate.

What was the relative increase in KCC2 expression compared with vehicle controls?

Relative increase in KCC2 expression vs vehicle controls was slightly higher than 4-fold, see Suppl Fig. 1B.

The authors note a Z'-factor of 0.94

When recalculating the Z'-factor we obtained a value of 0.73. We apologize for the erroneous determination of the Z-factor previously, it was caused by a spread-sheet error which we regret. However, 0.73 remains a strong Z'-factor, bespeaking of the robustness of our primary screening assay.

How many wells were used to calculate the Z-factor?

Thirty experiments were used for the calculation of the Z'-factor.

Was the Z-factor calculated for each plate?

It was not calculated for each plate given the small number of wells dedicated to controls per well. We felt that striking a balance between controls that we carry along vs number of compounds we can analyze, we had to maximize the number of compounds screened, given that the screen was conducted in primary cortical neurons. We are confident about the consistency and robust nature of the positive effects of a four-fold increase of positive control over vehicle control.

4) Full KP dose response experiments traversing multiple logs should be performed for luciferase, and ideally mRNA levels, at least in primary mouse cortical neurons, to establish an EC50.

We now show these data in our new Suppl Fig. 1c-d. Response to multiple doses of KP at the *Kcc2/KCC2* mRNA level is shown in Fig. 2.

In addition, inclusion of similar data for the other 3 hits identified would strengthen the manuscript.

We have tested two of the three other hit compounds and found they also increased *Kcc2* mRNA expression in a dose-dependent manner, approximately with similar potency to KP, shown now in Suppl Fig. 1e. - The third compound could not be purchased or retrieved for our use.

5) The KCC2 and synaptophysin co-stainings (Fig. 2E-F) are overinterpreted. Why is synaptophysin so weak? Is the culture very immature? Are the authors proposing that KP induces maturation, and that its effect on *Kcc2* transcription is indirect? An analysis of the effects of KP on primary human neurons that are cultured until they are mature enough to express synaptophysin, should be performed, as it would better address the question of

whether KP is increasing KCC2 in human neurons in a more relevant context. In addition, other markers of synaptic maturation need to be included to support the claim on p.6 that “in human primary cortical neurons we found enhanced synaptic maturation and increased KCC2 expression, most likely causally linked”.

We appreciate the reviewer’s comment regarding the overinterpretation of our results; we have now revised the text accordingly.

Primary human neuronal cultures used in this study were derived from fetal cortices at the stages (15 – 20 weeks of gestation) when neither KCC2 nor synaptophysin cortical expressions are fully developed. These short-term cultures (fixed at 10 days *in vitro* after 4 days of KP treatment) were used to test whether KP affects KCC2 expression in human neurons (as it does in mouse neurons). The results showed (by both RT-qPCR and ICC) that KP significantly increased the KCC2 levels in the human fetal cortical cultures. In addition, the level of synaptophysin labeling was increased, consistent with a general acceleration of neuronal differentiation/maturation. We edited the corresponding figure legend and main text (Results section) to clarify this point. We did not investigate the causal relationships between these two phenomena (KCC2 and synaptophysin both increase in response to KP) as it was beyond the scope of our study, which is specifically focused on the effect of KP on KCC2 gene expression. Previous studies that suggest an interrelationship between increasing KCC2 expression and increasing neuronal differentiation can be referenced here (Cell 105, 521-532, 2001; J Neurosci 29, 14652-14662, 2009). KP may regulate neuronal differentiation and synapse formation via inhibition of GSK3 β -dependent pathways (PNAS 2011, 108, 379-384, 2011; J Biol Chem 288, 9634-9647, 2013). We agree with the reviewer that it would be of interest to extend the current study and fully characterize the relationships between the dynamics of synaptophysin expression/localization, KCC2 appearance and the maturation of synapses in human fetal neuronal cultures at different stages of differentiation. We plan to perform these studies in the future.

6) The increased nuclear cytoplasmic localization shown in Fig. 6C and D is not clear from the ICC images. How was the quantification performed? Is it simply the average intensity of nuclear expression? A ratio of nuclear to cytoplasmic expression levels should be included. This would address whether the ratio has increased or whether overall levels of delta catenin have increased, and also allow for normalization between cultures.

We have addressed a closely related question in response to Reviewer 1.

"Quantification was performed on confocal micrographs so that immunoreactivity in the nucleus was recorded strictly separate from cytoplasmic immunoreactivity. Nuclear immunoreactivity was measured as relative value, normalized for cytoplasmic immunoreactivity. We measured cytoplasmic immunoreactivity and it remained unchanged between groups.

This method enabled us to draw conclusions about abundance of δ -cat in the nucleus in response to KP treatment."

7) There is a glancing reference to a similar screen that was performed in the context of Rett syndrome (Tang, X. et al., 2019). One of the confirmed KCC2 inducing hits that was investigated in this study was the GSK3b inhibitor BIO, in addition to follow-up with the GSK3b inhibitor TWS-119. This reference deserves more discussion as it considerably strengthens the authors’ identification of a GSK3b inhibitor, KP, that increases Kcc2 expression.

We gladly take the suggestion of the reviewer and comment more extensively on the pioneering study of Tang-et-al.

Minor comments:

1) The authors have included a large amount of experimental data, so much so that there are many points throughout the manuscript where the reader needs to search for basic information (e.g. days of treatment), making the manuscript difficult to read. A careful editing of the results section to ensure inclusion of information that the reader requires to assess the results should be performed. If space constraint is a problem, the description of the screen could easily be moved to the supplementary data section.

We have taken the reader’s comment to heart and thoroughly revised our ms with a single aim of improved clarity as a result of careful editing.

As we state in response to Reviewer 1: "We hope that our additional experimentation and increased

clarity of write-up, guided by the constructive critique of the reviewers, now anchor our key conclusion more firmly."

2) Validation of KCC2 antibodies is unclear on p. 42 "Anti-KCC2 primary antibodies were validated with developing rat primary cortical neurons; we observed an increase in staining pattern that tightly matched increase of Kcc2 mRNA expression." Based on what assay? ICC? Western?

As raised by Reviewer 1 relating to Fig. 4, we have answered this question.

We have conducted additional experiments that validate the antibody, shown in Suppl. Fig. 2.

3) The manuscript would be strengthened by including a discussion of whether there are GSK3b inhibitors in clinical development or trials.

We have included this discussion now, as thoughtfully suggested by the Reviewer.

Two clinical trials in neurodegenerative diseases are listed in clinicaltrials.gov, about a decade ago, without results. We conclude that the GSK3 β inhibitory molecules were not effective. In terms of pipeline development, we notice that British AMO Pharma has tideglusib which is in phase-II clinical trials for congenital myotonic dystrophy, a rare condition. If this drug could come to market, a repurposing trial in a chronic pain condition such as chemotherapy induced peripheral neuropathy (CIPN) or painful diabetic neuropathy could be run straightforwardly.

Kenpaullone has never been used in a clinical trial and is currently not under advanced clinical development, as a result of our inquiry, including *personal communication* Kunick & Meijer (2020).

4) All supplementary figure references need to be edited. For example, there is no supp Fig. S1 or S5, and there is inconsistent naming (Suppl Fig. vs Fig. S), etc., too many instances to list here.

We have addressed this in response to Reviewer1 and changed our numbering in the revised ms.

REVIEWER COMMENTS

Reviewer #1 (Remarks to the Author):

In this paper, Yeo et al. reported that kenpaullone (KP), an inhibitor of the GSK3 pathway, has antinociceptive properties in different preclinical models of chronic pain. They showed that KP-mediated analgesia occurs via upregulation of KCC2 and concurrent increased chloride extrusion from neurons in the dorsal horn of the spinal cord. Overall, the authors have addressed most of my concerns. The study is comprehensive and interesting, and the added experiments in the revised manuscripts corroborate and strengthen the author's conclusions. The addition of more methodological details will allow now others to replicate the study if need be.

A critical point remains though: Although Fig 4B is now of better quality, an inset showing with a higher magnification of the staining should be added. Furthermore, it is necessary to show the contralateral spinal cord from the same mice as this is critical to evaluate if the decrease in the ipsi side is real. It follows that the contralateral side must be used to quantify the decrease. Different animals will have different levels of Kcc2 expression and therefore ipsi vs ipsi across animals is meaningless. The same observation applies to the renormalization experiment in which ipsi has to be compared to the contra side; not to an ipsi side of a different animal which may have had a different baseline level to start with.

Minor:

The authors mention that the effect of KP was observed throughout the spinal cord but only analyzed laminae I-II because they are the critical layers for pain transmission to higher brain centers. However, I still think that this is an important piece of data that should be at the very least mentioned in the text and discussed as laminae I-II are not the sole critical layers of the spinal cord. Projection neurons exist in all laminae (except lamina II) and therefore contribute to the overall behavioral effect.

Reviewer #3 (Remarks to the Author):

Attachment provided with replies to authors responses to initial review

Reviewer #3 (Remarks to the Author):

Based on the hypothesis that restoration of Kcc2 expression levels and improvement of GABA evoked chloride reversal potentials will relieve pathologic pain, Liedtke and colleagues describe a screen for small molecules that increase Kcc2 transcription using primary mouse cortical neurons harboring a Kcc2 promoter luciferase reporter. The screen was performed in 24 well plates against 1057 growth regulating compounds from collections curated by NCI, based on the idea that many of them likely impact transcriptional and epigenetic mechanisms. Four hit compounds were identified based on luciferase activity, Kcc2 transcript levels, and measurements of chloride ion concentration. One of those, kenpaullone (KP), was selected for further analyses. First the authors show that KP treatment of primary cortical neurons leads to dose responsive increases in Kcc2 transcripts by RT-PCR, and KCC2 protein by ICC. Moreover, they show that the KCC2 chloride transport inhibitor VU0240551, blocks the effect of KP on [Cl⁻], suggesting it is KCC2 dependent. Some of these results are confirmed in primary human neurons where they show dose dependent Kcc2 transcript increases, as well as KCC2 protein increase by ICC that correlates with increased synaptophysin expression. They then go on to show that in vivo, using two pain models in mice-nerve constriction injury, and bone cancer-that KP acts as an analgesic for mechanical withdrawal and allodynia, again in a KCC2 dependent manner based on co-administration of VU0240551. These in vivo studies are followed up with an ex-vivo examination of dissected spinal cord dorsal horn (SCDH), where increased Kcc2 transcription by RT-PCR, and KCC2 protein by ICC is demonstrated. In addition, perforated patch recordings were performed on slices to demonstrate normalization of EGABA in KP treated animals. Finally, the authors perform a series of experiments, including Kcc2 promoter crunching, aimed at understanding the mechanism of action of KP. These experiments lead to the proposal that KP inhibits GSK3b (not CDKs), resulting in decreased S276 phosphorylation, stabilization and nuclear translocation of delta catenin, and upregulation of Kcc2 via the delta catenin-Kaiso pathway. In a final in vivo experiment, the authors show that overexpression of S276 delta catenin in SCDH via AAV9, phenocopies KP treatment. This manuscript describes extensive testing of the hypothesis that regulation of Kcc2 transcript levels by a GSK3b dependent pathway represents a potential target for pain management, an important, unmet medical need. The manuscript is of general interest, especially to those in the field of pain management.

We appreciate the reviewer's positive evaluation.

However, in its current form the manuscript is not recommended for publication in Nature Communications. If the authors could address the concerns below, it is recommended that the manuscript be reconsidered.

We thank the reviewer for the constructive comments which we believe have greatly improved this study. We hope that our revised ms can now be reconsidered.

Major comments: 1) The authors are proposing a direct relationship between KP treatment and KCC2 protein levels. However, the manuscript is entirely dependent on ICC to establish that KP increases KCC2 protein levels. Western blots need to be included at least for primary mouse cortical neurons, to allow quantification of KCC2 protein levels over time in response to KP.

We have now determined abundance of KCC2 protein in neuronal cultures by conducting microfluidicelectrophoretic separation of protein extract from cultured rat primary neurons with subsequent immunodetection. This leads to a result very similar to Western blotting. The result shows a specific band of KCC2 at a molecular weight of approximately 150kDa, appearing there also in mouse brain protein extract, using an anti-KCC2 antibody against the N-terminus which has previously been validated. We demonstrate significant upregulation of KCC2 in response to KP treatment (0.5µM). These results are now included into revised Fig. 2 and Suppl Fig. 2.

Results should be normalized to loading control (beta tubulin). Quantified values should be noted in the text or on the graph, as the sample size is small and the values are relatively close, such that they do not appear significantly different. Normalization may help.

2) In addition, protein levels assessed by ICC in Fig. 2A should be quantified, and if possible, the spinal cord sections shown in Fig. 4B should include a control marker not affected by KP that can be used to normalize KCC2 expression between animals.

We have quantified ICC as shown now in Fig. 2b. Re Fig. 4b we did not find suitable control markers. We replaced the images of Fig. 4b with higher resolution. We also validated the specificity of KCC2 antibody

(Suppl Fig. 2a), also see response to R1.

For Fig. 4b There aren't any markers which could be used to normalize KCC2 expression between animals? Neuronal markers? The authors should note in the text the caveat that the quantifications were not normalized.

3) More information needs to be provided regarding quality control for the screen that will allow the reader to assess assay robustness. What was the coefficient of variation across wells?

We calculated the coefficient of variation as 6% for vehicle control and 5.3% for positive control.

A scatter plot of the results including vehicle controls should be presented, ideally as Z-scores, or percent change from vehicle controls, so inter-plate results could be compared.

We now show a scatter plot of 30 experiments in Suppl Fig. 1a-b. These 30 experiments were the primary screen. We show the reporter gene readout of the vehicle control vs the positive control, and are also showing it as -fold increase of positive control vs vehicle control in a dot-plot. Importantly, our screen was conducted using primary cortical neurons derived from late rat embryos. With this type of cells for a screen, their source and resulting batch-to-batch (that is rat pregnancy to rat pregnancy) variation it is worth remembering. Per screening experiment we screened between 1-3 plates, depending on cell yield/availability. We know that maturation of neurons as they are taken into culture from individual pregnancies can be variable, therefore we believe that our efforts at accomplishing consistent plate-to-plate drive on the *Kcc2* promoter have yielded quite satisfactory outcome, when looking at the raw RLU output. Number of wells per control condition were between 2 and 4 per plate.

The scatter plot, which the authors have included in the revised version, is not useful- it must include, as requested, the results from the screen, not just the controls.

As the authors correctly point out there is an expectation of plate-to-plate variability, especially because of batch effects. Thus, to normalize the data points from each plate (or "experiment" they can be converted to Z-scores (or perhaps percent activity compared with TSA). The results from all 30 experiments can then be plotted on the same scatter plot and the reader can then assess the results from the entire screen and get a better understanding of how the 137 hits were picked. Based on the consistency of the raw RLU output presented for the controls, such a scatter plot with the screening data points should be informative.

What was the relative increase in KCC2 expression compared with vehicle controls?

Relative increase in KCC2 expression vs vehicle controls was slightly higher than 4-fold, see Suppl Fig. 1B.

The authors note a Z'-factor of 0.94

When recalculating the Z'-factor we obtained a value of 0.73. We apologize for the erroneous determination of the Z-factor previously, it was caused by a spread-sheet error which we regret.

However, 0.73 remains a strong Z'-factor, bespeaking of the robustness of our primary screening assay.

How many wells were used to calculate the Z-factor?

Thirty experiments were used for the calculation of the Z'-factor.

This is an inappropriate use of Z-factor- it should not be determined by combining data points from independent experiments and averaging over many independent runs of the assay on different batches of cells. The Z-factor should be calculated independently for each plate (or "experiment" of 1 to 3 plates in this case), or it should be removed, as the screen performed in this study was low throughput. Alternatively, if there are data sets from assay development efforts where the number of wells was significantly large to calculate a Z-factor with confidence, then those numbers could be reported for each single plate where it was calculated. The Z-factor is used in assay development as an indication that the window separating vehicle and control compound is large enough to screen either as singletons ($Z > 0.5$) or in duplicate ($Z > 0$). It is then typically used for screening campaigns for every plate as an internal control to give confidence in hit picking

for each plate.

Was the Z-factor calculated for each plate?

It was not calculated for each plate given the small number of wells dedicated to controls per well. We felt that striking a balance between controls that we carry along vs number of compounds we can analyze, we had to maximize the number of compounds screened, given that the screen was conducted in primary cortical neurons. We are confident about the consistency and robust nature of the positive effects of a four-fold increase of positive control over vehicle control.

4) Full KP dose response experiments traversing multiple logs should be performed for luciferase, and ideally mRNA levels, at least in primary mouse cortical neurons, to establish an EC50.

We now show these data in our new Suppl Fig. 1c-d. Response to multiple doses of KP at the *Kcc2/KCC2* mRNA level is shown in Fig. 2.

In addition, inclusion of similar data for the other 3 hits identified would strengthen the manuscript.

We have tested two of the three other hit compounds and found they also increased *Kcc2* mRNA expression in a dose-dependent manner, approximately with similar potency to KP, shown now in Suppl Fig. 1e. -The third compound could not be purchased or retrieved for our use.

5) The KCC2 and synaptophysin co-stainings (Fig. 2E-F) are overinterpreted. Why is synaptophysin so weak? Is the culture very immature? Are the authors proposing that KP induces maturation, and that its effect on *Kcc2* transcription is indirect? An analysis of the effects of KP on primary human neurons that are cultured until they are mature enough to express synaptophysin, should be performed, as it would better address the question of whether KP is increasing KCC2 in human neurons in a more relevant context. In addition, other markers of synaptic maturation need to be included to support the claim on p.6 that “in human primary cortical neurons we found enhanced synaptic maturation and increased KCC2 expression, most likely causally linked”.

We appreciate the reviewer’s comment regarding the overinterpretation of our results; we have now revised the text accordingly. Primary human neuronal cultures used in this study were derived from fetal cortices at the stages (15 –20 weeks of gestation) when neither KCC2 nor synaptophysin cortical expressions are fully developed. These short-term cultures (fixed at 10 days *in vitro* after 4 days of KP treatment) were used to test whether KP affects KCC2 expression in human neurons (as it does in mouse neurons). The results showed (by both RT-qPCR and ICC) that KP significantly increased the KCC2 levels in the human fetal cortical cultures. In addition, the level of synaptophysin labeling was increased, consistent with a general acceleration of neuronal differentiation/maturation. We edited the corresponding figure legend and main text (Results section) to clarify this point. We did not investigate the causal relationships between these two phenomena (KCC2 and synaptophysin both increase in response to KP) as it was beyond the scope of our study, which is specifically focused on the effect of KP on *KCC2* gene expression. Previous studies that suggest an interrelationship between increasing *KCC2* expression and increasing neuronal differentiation can be referenced here (Cell 105, 521-532, 2001; J Neurosci 29,14652-14662, 2009). KP may regulate neuronal differentiation and synapse formation via inhibition of GSK3b-dependent pathways (PNAS 2011, 108, 379-384, 2011; J Biol Chem 288, 9634-9647, 2013). We agree with the reviewer that it would be of interest to extend the current study and fully characterize the relationships between the dynamics of synaptophysin expression/localization, KCC2 appearance and the maturation of synapses in human fetal neuronal cultures at different stages of differentiation. We plan to perform these studies in the future.

The level of synaptophysin expression is so low as to be uninterpretable. Inclusion of this result raises more questions than it answers and it should be removed. Additional markers need to be included to better assess maturity of the culture- a few standard markers does not seem beyond the scope of this study, and is actually the norm when assessing neuronal cultures.

6) The increased nuclear cytoplasmic localization shown in Fig. 6C and D is not clear from the ICC images. How was the quantification performed? Is it simply the average intensity of nuclear expression? A ratio of nuclear to cytoplasmic expression levels should be included. This would address whether the ratio has increased or whether overall levels of delta catenin have increased, and also allow for normalization between cultures.

We have addressed a closely related question in response to Reviewer 1. **"Quantification was performed on confocal micrographs so that immunoreactivity in the nucleus was recorded strictly separate from cytoplasmic immunoreactivity. Nuclear immunoreactivity was measured as relative value, normalized for cytoplasmic immunoreactivity. We measured cytoplasmic immunoreactivity and it remained unchanged between groups. This method enabled us to draw conclusions about abundance of β -cat in the nucleus in response to KP treatment."** From the data shown it does not appear that the "cytoplasmic immunoreactivity remained unchanged". It appears that both nuclear and cytoplasmic are increased. A detailed description in the methods of how cytoplasmic and nuclear immunoreactivity were determined (e.g. per area of each compartment in each cell? sum intensity in each compartment per cell?), and how the nuclear reactivity was normalized is needed. For instance, the nuclear catenin IR relU for the vehicle on WT in Fig. 6D is ~ 1.0 . Does that mean that nuclear and cytoplasmic immunoreactivities have a 1:1 ratio and are of equivalent levels? For 6C vehicle it is ~ 0.4 . Does that mean that cytoplasmic: nuclear ratio is $\sim 2.5:1$?

The images do not indicate this.

7) There is a glancing reference to a similar screen that was performed in the context of Rett syndrome (Tang, X. et al., 2019). One of the confirmed KCC2 inducing hits that was investigated in this study was the GSK3b inhibitor BIO, in addition to follow-up with the GSK3b inhibitor TWS-119. This reference deserves more discussion as it considerably strengthens the authors' identification of a GSK3b inhibitor, KP, that increases Kcc2 expression. **We gladly take the suggestion of the reviewer and comment more extensively on the pioneering study of Tang-et-al.**

Minor comments: 1) The authors have included a large amount of experimental data, so much so that there are many points throughout the manuscript where the reader needs to search for basic information (e.g. days of treatment), making the manuscript difficult to read. A careful editing of the results section to ensure inclusion of information that the reader requires to assess the results should be performed. If space constraint is a problem, the description of the screen could easily be moved to the supplementary data section. **We have taken the reader's comment to heart and thoroughly revised our ms with a single aim of improved clarity as a result of careful editing. As we state in response to Reviewer 1: "We hope that our additional experimentation and increased clarity of write-up, guided by the constructive critique of the reviewers, now anchor our key conclusion more firmly."**
The manuscript remains difficult to read and requires further editing

2) Validation of KCC2 antibodies is unclear on p. 42 "Anti-KCC2 primary antibodies were validated with developing rat primary cortical neurons; we observed an increase in staining pattern that tightly matched increase of Kcc2 mRNA expression." Based on what assay? ICC? Western?

As raised by Reviewer 1 relating to Fig. 4, we have answered this question. We have conducted additional experiments that validate the antibody, shown in Suppl. Fig. 2.

3) The manuscript would be strengthened by including a discussion of whether there are GSK3b inhibitors in clinical development or trials.

We have included this discussion now, as thoughtfully suggested by the Reviewer. Two clinical trials in neurodegenerative diseases are listed in clinicaltrials.gov, about a decade ago, without results. We conclude that the GSK3 β inhibitory molecules were not effective. In terms of pipeline development, we notice that British AMO Pharma has tideglusib which is in phase-II clinical trials for congenital myotonic dystrophy, a rare condition. If this drug could come to market, a repurposing trial in a chronic pain condition such as chemotherapy induced peripheral neuropathy (CIPN) or painful diabetic neuropathy could be run straightforwardly. Kenpaullone has never been used in a clinical trial and is currently not under advanced clinical development, as a result of our inquiry, including *personal communication* Kunick & Meijer (2020).

4) All supplementary figure references need to be edited. For example, there is no supp Fig. S1 or S5, and there is inconsistent naming (Suppl Fig. vs Fig. S), etc., too many instances to list here.

We have addressed this in response to Reviewer1 and changed our numbering in the revised ms.

July 31, 2021

After discussion of remaining critical issues with our study, we have conducted additional experimentation and data analysis as suggested by the reviewers' critique, leading to a re-revised manuscript including modified figures and data presentation.

To our understanding and sentiment, the previous critique of our revised manuscript has been addressed, see our responses below. Hopefully our re-revised ms. can be reconsidered now for publication.

Reviewer #1 (Remarks to the Author):

In this paper, Yeo et al. reported that kenpaullone (KP), an inhibitor of the GSK3 pathway, has antinociceptive properties in different preclinical models of chronic pain. They showed that KP-mediated analgesia occurs via upregulation of KCC2 and concurrent increased chloride extrusion from neurons in the dorsal horn of the spinal cord. Overall, the authors have addressed most of my concerns. The study is comprehensive and interesting, and the added experiments in the revised manuscripts corroborate and strengthen the author's conclusions. The addition of more methodological details will allow now others to replicate the study if need be.

We thank the reviewer for this verdict after our extensive revision in response to her/his detailed first round critique.

A critical point remains though: Although Fig 4B is now of better quality, an inset showing with a higher magnification of the staining should be added. Furthermore, it is necessary to show the contralateral spinal cord from the same mice as this is critical to evaluate if the decrease in the ipsi side is real. It follows that the contralateral side must be used to quantify the decrease. Different animals will have different levels of Kcc2 expression and therefore ipsi vs ipsi across animals is meaningless. The same observation applies to the renormalization experiment in which ipsi has to be compared to the contra side; not to an ipsi side of a different animal which may have had a different baseline level to start with.

revised Figure 4b.

KP treatment increases KCC2 expression in the SCDH. Left-hand micrographs: representative KCC2 immuno-staining ipsi-lateral vs contralateral to injury (see antibody validation in Supplementary Fig. 2a) of the SCDH in nerve constriction injury (PSNL), region-of-interest for densitometric measurement of KCC2 in the SCDH outlined with dotted white line in upper-left micrograph, focus on Rexed layers I-II because of their relevance for neurotransmission of nociceptive afferent signals. KCC2 signals were normalized for Nissl stain (green fluorescent signal, inset micrographs, see Supplementary Fig. 4a), which did not differ between conditions and sides, see Supplementary Fig. 4a-b. Bar diagrams: KP (10mg/kg; daily treatment for 7d after PSNL) increases KCC2 protein expression in SCDH vs vehicle control in PSNL nerve constriction, ipsi-lateral to injury, no difference in KCC2 signal contralaterally. n=6-7 mice/group, *p<0.05; **p<0.01, one-way ANOVA.

We have taken to heart the insightful critique of the reviewer and have conducted this experiment.

Results, now shown in revised Fig. 4b (left-hand) and Supplementary Fig. 4a-c, are clear in that the ipsilateral, injured site shows a significant decrease of KCC2 protein expression in the spinal cord dorsal horn (SCDH). This defect, in keeping with decreased Kcc2 mRNA of microexcised SCDH tissue (Fig. 4a), is repaired in response to KP treatment.

Our revision experiments show: (i) KCC2 protein expression, normalized for Nissl, is significantly diminished ipsilateral to peripheral nerve constriction injury; (ii) KP treatment repairs this defect on the injured side. The contralateral side does not show a significant difference of KCC2,

normalized for Nissl, between injured vs injured plus KP treatment groups.

In addition, for KCC2 protein expression, we also have determined differences between contra- and ipsilateral, normalized for contralateral, and compared these metrics between vehicle-control treated mice vs. KP-treated animals. There is a significant difference between injury treated with vehicle-control group vs injury treated with KP group, shown in Supplementary Fig. 4c.

These new findings are represented in our revised Figure 4b (see also response to second reviewer), and in revised Suppl Fig. 4.

Minor:

The authors mention that the effect of KP was observed throughout the spinal cord but only analyzed laminae I-II because they are the critical layers for pain transmission to higher brain centers. However, I still think that this is an important piece of data that should be at the very least mentioned in the text and discussed as laminae I-II are not the sole critical layers of the spinal cord. Projection neurons exist in all laminae (except lamina II) and therefore contribute to the overall behavioral effect.

We have revised our manuscript by mentioning and appropriately commenting on the co-contributory role of deep layers, beyond superficial layers I-II in the SCDH, in pain transmission.

Reviewer #3 (Remarks to the Author) (most recent remarks in direct response to our previous answer):

Based on the hypothesis that restoration of Kcc2 expression levels and improvement of GABA evoked chloride reversal potentials will relieve pathologic pain, Liedtke and colleagues describe a screen for small molecules that increase Kcc2 transcription using primary mouse cortical neurons harboring a Kcc2 promoter luciferase reporter. The screen was performed in 24 well plates against 1057 growth regulating compounds from collections curated by NCI, based on the idea that many of them likely impact transcriptional and epigenetic mechanisms. Four hit compounds were identified based on luciferase activity, Kcc2 transcript levels, and measurements of chloride ion concentration. One of those, kenpauillone (KP), was selected for further analyses.

First the authors show that KP treatment of primary cortical neurons leads to dose responsive increases in Kcc2 transcripts by RT-PCR, and KCC2 protein by ICC. Moreover, they show that the KCC2 chloride transport inhibitor VU0240551, blocks the effect of KP on [Cl⁻], suggesting it is KCC2 dependent. Some of these results are confirmed in primary human neurons where they show dose dependent Kcc2 transcript increases, as well as KCC2 protein increase by ICC that correlates with increased synaptophysin expression. They then go on to show that in vivo, using two pain models in mice- nerve constriction injury, and bone cancer- that KP acts as an analgesic for mechanical withdrawal and allodynia, again in a KCC2 dependent manner based on co-administration of VU0240551. These in vivo studies are followed up with an ex-vivo examination of dissected spinal cord dorsal horn (SCDH), where increased Kcc2 transcription by RT-PCR, and KCC2 protein by ICC is demonstrated. In addition, perforated patch recordings were performed on slices to demonstrate normalization of EGABA in KP treated animals. Finally, the authors perform a series of experiments, including Kcc2 promoter crunching, aimed at understanding the mechanism of action of KP. These experiments lead to the proposal that KP inhibits GSK3b (not CDKs), resulting in decreased S276 phosphorylation, stabilization and nuclear translocation of delta catenin, and upregulation of Kcc2 via the delta catenin-Kaiso pathway. In a final in vivo experiment, the authors show that overexpression of S276 delta catenin in SCDH via AAV9, phenocopies KP treatment.

This manuscript describes extensive testing of the hypothesis that regulation of Kcc2 transcript levels by a GSK3b dependent pathway represents a potential target for pain management, an important, unmet medical need. The manuscript is of general interest, especially to those in the field of pain management.

We appreciate the reviewer's positive evaluation.

However, in its current form the manuscript is not recommended for publication in Nature Communications. If the authors could address the concerns below, it is recommended that the manuscript be reconsidered.

We thank the reviewer for the constructive comments which we believe have greatly improved this study. We hope that our revised ms can now be reconsidered.

Major comments:

1) The authors are proposing a direct relationship between KP treatment and KCC2 protein levels. However, the manuscript is entirely dependent on ICC to establish that KP increases KCC2 protein levels. Western blots need to be included at least for primary mouse cortical neurons, to allow quantification of KCC2 protein levels over time in response to KP.

We have now determined abundance of KCC2 protein in neuronal cultures by conducting microfluidic electrophoretic separation of protein extract from cultured rat primary neurons with subsequent immunodetection. This leads to a result very similar to Western blotting. The result shows a specific band of KCC2 at a molecular weight of approximately 150kDa, appearing there also in mouse brain protein extract, using an anti-KCC2 antibody against the N-terminus which has previously been validated. We demonstrate significant upregulation of KCC2 in response to KP treatment (0.5µM). These results are now included into revised Fig. 2 and Suppl Fig. 2.

Results should be normalized to loading control (beta tubulin). Quantified values should be noted in the text or on the graph, as the sample size is small and the values are relatively close, such that they do not appear significantly different. Normalization may help.

We did normalize to loading control, neuronal β_{III}-tubulin, see Suppl Fig. 2b-c. Measurements are significantly different, and metrical values are indicated in Suppl Fig. 2c.

2) In addition, protein levels assessed by ICC in Fig. 2A should be quantified, and if possible, the spinal cord sections shown in Fig. 4B should include a control marker not affected by KP that can be used to normalize

KCC2 expression between animals.

We have quantified ICC as shown now in Fig. 2b.

Re Fig. 4b we did not find suitable control markers. We replaced the images of Fig. 4b with higher resolution. We also validated the specificity of KCC2 antibody (Suppl Fig. 2a), also see response to R1. For Fig. 4b There aren't any markers which could be used to normalize KCC2 expression between animals? Neuronal markers? The authors should note in the text the caveat that the quantifications were not normalized.

We have now conducted Nissl staining in our revision experiments, when immunolabeling the spinal cord for KCC2. We have previously used this method with benefit on SCDH (Jiang et al., iScience 23, 101570, 2020). Results are shown in revised Fig. 4b, and Suppl Fig. 4a-b, showing both, micrographs and their quantification. The Nissl stain shows no significant difference between any condition or side of the spinal cord. We measured expression for the Nissl stain and used these metrics for normalization of KCC2 protein expression.

Our revision experiments show: (i) KCC2 protein expression, normalized for Nissl, is significantly diminished ipsilateral to peripheral nerve constriction injury; (ii) KP treatment repairs this defect on the injured side. The contralateral side does not show a significant difference of KCC2, normalized for Nissl, between injured treated with vehicle control group vs injured with KP treatment group.

Please also see our response to R1 re Fig. 4b.

3) More information needs to be provided regarding quality control for the screen that will allow the reader to assess assay robustness.

What was the coefficient of variation across wells?

We calculated the coefficient of variation as 6% for vehicle control and 5.3% for positive control.

A scatter plot of the results including vehicle controls should be presented, ideally as Z-scores, or percent change from vehicle controls, so inter-plate results could be compared.

We now show a scatter plot of 30 experiments in Suppl Fig. 1a-b. These 30 experiments were the primary screen. We show the reporter gene readout of the vehicle control vs the positive control, and are also showing it as -fold increase of positive control vs vehicle control in a dot-plot.

Importantly, our screen was conducted using primary cortical neurons derived from late rat embryos. With this type of cells for a screen, their source and resulting batch-to-batch (that is rat pregnancy to rat pregnancy) variation it is worth remembering. Per screening experiment we screened between 1-3 plates, depending on cell yield/availability. We know that maturation of neurons as they are taken into culture from individual pregnancies can be variable, therefore we believe that our efforts at accomplishing consistent plate-to-plate drive on the *Kcc2* promoter have yielded quite satisfactory outcome, when looking at the raw RLU output. Number of wells per control condition were between 2 and 4 per plate.

The scatter plot, which the authors have included in the revised version, is not useful- it must include, as requested, the results from the screen, not just the controls.

As the authors correctly point out there is an expectation of plate-to-plate variability, especially because of batch effects. Thus, to normalize the data points from each plate (or "experiment" they can be converted to Z-scores (or perhaps percent activity compared with TSA). The results from all 30 experiments can then be plotted on the same scatter plot and the reader can then assess the results from the entire screen and get a better understanding of how the 137 hits were picked. Based on the consistency of the raw RLU output presented for the controls, such a scatter plot with the screening data points should be informative.

We have now included into the revised scatter plot (Suppl Fig. 1b) the results from the screen, and show both metrics for all 1057 compounds as well as for top-137 compounds emanating as "winners" from 1st round.

What was the relative increase in KCC2 expression compared with vehicle controls?

Relative increase in KCC2 expression vs vehicle controls was slightly higher than 4-fold, see Suppl Fig. 1B.

The authors note a Z'-factor of 0.94

When recalculating the Z'-factor we obtained a value of 0.73. We apologize for the erroneous determination of the Z-factor previously, it was caused by a spread-sheet error which we regret.

However, 0.73 remains a strong Z'-factor, bespeaking of the robustness of our primary screening assay.

How many wells were used to calculate the Z-factor?

Thirty experiments were used for the calculation of the Z'-factor.

This is an inappropriate use of Z-factor- it should not be determined by combining data points from independent experiments and averaging over many independent runs of the assay on different batches of cells. The Z-factor should be calculated independently for each plate (or "experiment" of 1 to 3 plates in this case), or it should be removed, as the screen performed in this study was low throughput. Alternatively, if there are data sets from assay development efforts where the number of wells was significantly large to calculate a Z-factor with confidence, then those numbers could be reported for each single plate where it was calculated. The Z-factor is used in assay development as an indication that the window separating vehicle and control compound is large enough to screen either as singletons ($Z > 0.5$) or in duplicate ($Z > 0$). It is then typically used for screening campaigns for every plate as an internal control to give confidence in hit picking for each plate.

Considering the reviewer's thoughtful argument, we have decided to follow the reviewer's suggestion to remove the Z'-factor because we have conducted a non-typical screen at low throughput in primary neurons.

We hope that data provided and presented in our re-revised ms (Fig. 1, Suppl Fig. 1, Suppl Dataset 1) can now generate sufficient confidence.

Was the Z-factor calculated for each plate?

It was not calculated for each plate given the small number of wells dedicated to controls per well.

We felt that striking a balance between controls that we carry along vs number of compounds we can analyze, we had to maximize the number of compounds screened, given that the screen was conducted in primary cortical neurons. We are confident about the consistency and robust nature of the positive effects of a four-fold increase of positive control over vehicle control.

4) Full KP dose response experiments traversing multiple logs should be performed for luciferase, and ideally mRNA levels, at least in primary mouse cortical neurons, to establish an EC50.

We now show these data in our new Suppl Fig. 1c-d. Response to multiple doses of KP at the *Kcc2/KCC2* mRNA level is shown in Fig. 2.

In addition, inclusion of similar data for the other 3 hits identified would strengthen the manuscript.

We have tested two of the three other hit compounds and found they also increased *Kcc2* mRNA expression in a dose-dependent manner, approximately with similar potency to KP, shown now in Suppl Fig. 1e. - The third compound could not be purchased or retrieved for our use.

5) The KCC2 and synaptophysin co-stainings (Fig. 2E-F) are overinterpreted. Why is synaptophysin so weak? Is the culture very immature? Are the authors proposing that KP induces maturation, and that its effect on *Kcc2* transcription is indirect? An analysis of the effects of KP on primary human neurons that are cultured until they are mature enough to express synaptophysin, should be performed, as it would better address the question of whether KP is increasing KCC2 in human neurons in a more relevant context. In addition, other markers of synaptic maturation need to be included to support the claim on p.6 that "in human primary cortical neurons we found enhanced synaptic maturation and increased KCC2 expression, most likely causally linked".

We appreciate the reviewer's comment regarding the overinterpretation of our results; we have now revised the text accordingly.

Primary human neuronal cultures used in this study were derived from fetal cortices at the stages (15 – 20 weeks of gestation) when neither KCC2 nor synaptophysin cortical expressions are fully developed. These short-term cultures (fixed at 10 days *in vitro* after 4 days of KP treatment) were used to test whether KP affects KCC2 expression in human neurons (as it does in mouse neurons). The results showed (by both RT-qPCR and ICC) that KP significantly increased the KCC2 levels in the human fetal cortical cultures. In addition, the level of synaptophysin labeling was increased, consistent with a general acceleration of neuronal differentiation/maturation. We edited the corresponding figure legend and main text (Results section) to clarify this point. We did not investigate the causal relationships between these two phenomena (KCC2 and synaptophysin both increase in response to KP) as it was

beyond the scope of our study, which is specifically focused on the effect of KP on *KCC2* gene expression. Previous studies that suggest an interrelationship between increasing *KCC2* expression and increasing neuronal differentiation can be referenced here (Cell 105, 521-532, 2001; J Neurosci 29, 14652-14662, 2009). KP may regulate neuronal differentiation and synapse formation via inhibition of GSK3 β -dependent pathways (PNAS 2011, 108, 379-384, 2011; J Biol Chem 288, 9634-9647, 2013).

We agree with the reviewer that it would be of interest to extend the current study and fully characterize the relationships between the dynamics of synaptophysin expression/localization, *KCC2* appearance and the maturation of synapses in human fetal neuronal cultures at different stages of differentiation. We plan to perform these studies in the future.

The level of synaptophysin expression is so low as to be uninterpretable. Inclusion of this result raises more questions than it answers and it should be removed. Additional markers need to be included to better assess maturity of the culture- a few standard markers does not seem beyond the scope of this study, and is actually the norm when assessing neuronal cultures.

The level of synaptophysin expression is appreciable in our human fetal cortical cultures, also based on previous experience with synaptophysin expression in non-related experiments. We regret that the previous presentation evoked this impression.

We have revised the ms as follows in response to the reviewer.

revised Supplementary Figure 2g.

Primary human fetal cortical neurons; treatment with 0.4 μ M KP, as in main manuscript Figure 2f. Representative confocal images at DIV10 co-immunolabelled for *KCC2* (red) and synaptophysin (green), also shown is the bright-field image (left-hand) and nuclear stain (DAPI - blue fluorescence), right-hand side showing merged fluorescent micrographs. Note enhanced expression of synaptophysin and of *KCC2* in response to KP, also co-localization of *KCC2* and synaptophysin. Scalebar=10 μ m.

In the main manuscript Fig. 2g, see left-hand, we now show an updated micrograph of *KCC2* protein expression in the cultured human fetal neurons, plus the quantification (Fig. 2h). This refers to the main point of the study: treatment with KP evokes a similar effect in human primary neurons as it does in mouse and rat. This is particularly relevant in a translational medical context. We have changed our mentioning of this result and commenting on it in the re-revised ms to make this important issue more clear. Re synaptophysin staining, we are including revised micrographs in Supplementary Fig. 2g that hopefully do better justice to the expression levels

observed and measured (Supplementary Fig. 2h), and make more clear the co-localization with *KCC2*.

We hope that our redacted presentation can now address the critique of the reviewer, and that our overriding goal of including data on primary human fetal cortical neurons showing increased *KCC2* gene expression in response to KP has become more clear.

6) The increased nuclear cytoplasmic localization shown in Fig. 6C and D is not clear from the ICC images. How was the quantification performed? Is it simply the average intensity of nuclear expression? A ratio of nuclear to cytoplasmic expression levels should be included. This would address whether the ratio has increased or whether overall levels of delta catenin have increased, and also allow for normalization between cultures.

We have addressed a closely related question in response to Reviewer 1.

"Quantification was performed on confocal micrographs so that immunoreactivity in the nucleus was recorded strictly separate from cytoplasmic immunoreactivity. Nuclear immunoreactivity was measured as relative value, normalized for cytoplasmic immunoreactivity. We measured cytoplasmic immunoreactivity and it remained unchanged between groups.

This method enabled us to draw conclusions about abundance of δ -cat in the nucleus in response to KP

treatment."

From the data shown it does not appear that the "cytoplasmic immunoreactivity remained unchanged". It appears that both nuclear and cytoplasmic are increased.

We regret that we have not made this important issue more clear.

A detailed description in the methods of how cytoplasmic and nuclear immunoreactivity were determined (e.g. per area of each compartment in each cell? sum intensity in each compartment per cell?), and how the nuclear reactivity was normalized is needed.

Densitometry for δ -cat was determined for soma plus proximal neural process (in case of presence of such a process; $\leq 10\mu\text{m}$ distance from soma), excluding nuclear compartment, vs nucleus. These regions were marked in ImageJ, then their average density, background subtracted, was measured in ImageJ. Per neuron, nuclear density was divided by cytoplasmic density.

To illustrate, we are including an example of 2 primary neurons, from Fig. 6c.

a – nuclear compartment
b – soma and proximal neural process

a – 25.39
b – 38.96
0.65

a – 42.16
b – 32.96
1.28

The results, using the above method applied to each neuron is 1.28 for the KP-treated neuron vs 0.65 for the vehicle-treated neuron.

In Fig. 6c, each dot represents the metrics of this ratio for one neuron, with mean+SEM, illustrated by bar and error bar.

For instance, the nuclear catenin IR reIU for the vehicle on WT in Fig. 6D is ~ 1.0 .

For Fig. 6d, using the neuronalized cell line N2a, and two variables: +/- KP-treatment, S276A hu- δ -cat mutation or hu- δ -cat WT transgene, the mean of the subgroup " δ -cat(WT) + vehicle treatment" was set to "1" to allow a relative comparison. This factor was then applied to multiply the measured values for each cell for all four groups. The result is shown as bar diagram with individual data points, +SEM, level of statistically significant differences calculated with GraphPad Prism9.2, mixed model statistics.

Does that mean that nuclear and cytoplasmic immunoreactivities have a 1:1 ratio and are of equivalent levels?

No, please see above explanation - the factor used to set the mean of this group to "1" has to be taken into account. The relevant ratio is nuclear metrics divided by cytoplasmic.

For 6C vehicle it is ~ 0.4 . Does that mean that cytoplasmic: nuclear ratio is $\sim 2.5:1$?

That is correct, 0.4 translates to nuclear to cytoplasmic ratio of 1:2.5.

The images do not indicate this.

We hope this issue is more clear now with the above explanations.

We have tried to improve on this issue in our re-revised ms by providing a new schematic illustration in Suppl Fig. 6b, based on the above explanatory figure.

We have also included the requested descriptions of method in more detail in the Suppl Methods section of our re-revised paper.

7) There is a glancing reference to a similar screen that was performed in the context of Rett syndrome (Tang, X. et al., 2019). One of the confirmed KCC2 inducing hits that was investigated in this study was the GSK3b inhibitor BIO, in addition to follow-up with the GSK3b inhibitor TWS-119. This reference deserves more discussion as it considerably strengthens the authors' identification of a GSK3b inhibitor, KP, that increases Kcc2 expression.

We gladly take the suggestion of the reviewer and comment more extensively on the pioneering study of Tang-et-al.

Minor comments:

1) The authors have included a large amount of experimental data, so much so that there are many points throughout the manuscript where the reader needs to search for basic information (e.g. days of treatment), making the manuscript difficult to read. A careful editing of the results section to ensure inclusion of information that the reader requires to assess the results should be performed. If space constraint is a problem, the description of the screen could easily be moved to the supplementary data section.

We have taken the reader's comment to heart and thoroughly revised our ms with a single aim of improved clarity as a result of careful editing.

As we state in response to Reviewer 1: "We hope that our additional experimentation and increased clarity of write-up, guided by the constructive critique of the reviewers, now anchor our key conclusion more firmly."

The manuscript remains difficult to read and requires further editing

We have now involved a professional science writer and editor for language, syntax and style revisions, Dr David Hauss (Regeneron), whom we now acknowledge for his assistance. - We have not highlighted his language editorial changes in the "highlight-changes" version of our re-revised ms..

2) Validation of KCC2 antibodies is unclear on p. 42 "Anti-KCC2 primary antibodies were validated with developing rat primary cortical neurons; we observed an increase in staining pattern that tightly matched increase of Kcc2 mRNA expression." Based on what assay? ICC? Western?

As raised by Reviewer 1 relating to Fig. 4, we have answered this question.

We have conducted additional experiments that validate the antibody, shown in Suppl. Fig. 2.

3) The manuscript would be strengthened by including a discussion of whether there are GSK3b inhibitors in clinical development or trials.

We have included this discussion now, as thoughtfully suggested by the Reviewer.

Two clinical trials in neurodegenerative diseases are listed in clinicaltrials.gov, about a decade ago, without results. We conclude that the GSK3 β inhibitory molecules were not effective. In terms of pipeline development, we notice that British AMO Pharma has tideglusib which is in phase-II clinical trials for congenital myotonic dystrophy, a rare condition. If this drug could come to market, a repurposing trial in a chronic pain condition such as chemotherapy induced peripheral neuropathy (CIPN) or painful diabetic neuropathy could be run straightforwardly.

Kenpaullone has never been used in a clinical trial and is currently not under advanced clinical development, as a result of our inquiry, including *personal communication* Kunick & Meijer (2020).

4) All supplementary figure references need to be edited. For example, there is no supp Fig. S1 or S5, and there is inconsistent naming (Suppl Fig. vs Fig. S), etc., too many instances to list here.

We have addressed this in response to Reviewer1 and changed our numbering in the revised ms.

REVIEWER COMMENTS

Reviewer #1 (Remarks to the Author):

The authors have adequately addressed my concerns and the manuscript is now suitable for publication.